# Explanation-based Data Augmentation for Image Classification

**Sandareka Wickramanayake**   **Mong Li Lee**   **Wynne Hsu**
School of Computing
National University of Singapore
`{sandaw, leeml, whsu}@comp.nus.edu.sg`

## Abstract

Existing works have generated explanations for deep neural network decisions to provide insights into model behavior. We observe that these explanations can also be used to identify concepts that caused misclassifications. This allows us to understand the possible limitations of the dataset used to train the model, particularly the under-represented regions in the dataset. This work proposes a framework that utilizes concept-based explanations to automatically augment the dataset with new images that can cover these under-represented regions to improve the model performance. The framework is able to use the explanations generated by both interpretable classifiers and post-hoc explanations from black-box classifiers. Experiment results demonstrate that the proposed approach improves the accuracy of classifiers compared to state-of-the-art augmentation strategies.

## 1 Introduction

Machine learning models learn decision boundaries based on the training dataset. Ideally, a training dataset should provide sufficient variations in each class so that the model can learn the correct decision boundaries. However, when there are under-represented regions in the training dataset, the model will learn sub-optimal decision boundaries, leading to misclassification of data points [1, 2]. One way to address this problem is to augment the training dataset. A common data augmentation approach is to apply different transformations to the training samples [3] or to mix existing samples [4, 5]. However, such data augmentations only explore the neighborhood of these samples and may not cover the under-represented regions. Another approach is to use annotations such as part landmarks to obtain images from image resources such as Flickr or Google [6]. This approach may introduce out-of-distribution images and degrade the classifier performance.

Identifying the appropriate samples to add to the training dataset is crucial. Here, we propose to use model decision explanations to facilitate the search for samples from image repositories and augment the training dataset with samples from the under-represented regions to enable the model to learn the correct decision boundaries. Figure 1a shows the image of a juvenile *Black Tern* that has been misclassified as *White Breasted Nuthatch*. A closer examination of the training dataset reveals that it contains very few images of juvenile *Black Terns* (5 out of 30 images). These juvenile birds have white heads with black skullcap and black earmark, unlike the adult *Black Terns* [7] (see Figure 1b). In fact, the juvenile *Black Tern* in Figure 1a is more similar to *White Breasted Nuthatch* as shown in Figure 1c, leading to the misclassification. Clearly, there is a need to augment the training data to include more images of juvenile *Black Tern* in order to learn the correct decision boundary between *Black Terns* and *White Breasted Nuthatch*.

Methods to explain model decisions include pixel-level attributions [8, 9, 10], and high-level semantic concepts such as coarse-grained saliency maps [11, 12], interpretable basis decomposition [13], linguistic explanations [14, 15] and prototypes [16]. The work in [17] designs a Comprehensive

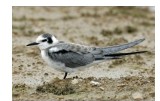 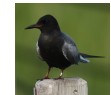 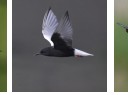 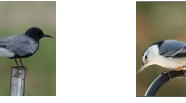 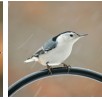 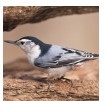

(a) Juvenile black tern.    (b) Images of adult black tern.    (c) Images of white breasted nuthatch.

Figure 1: Example of juvenile black tern that is misclassified as white breasted nuthatch.

Convolutional Neural Network called CCNN that explains the model decision in terms of word phrases, e.g., the bird in Figure 1a is classified as *White Breasted Nuthatch* due to the concepts *'white breast', 'white belly', 'long beak'*.

This paper introduces a framework called BRACE (BetteR Accuracy from Concept-based Explanation), which utilizes concept-based explanations to identify candidate samples from image repositories for data augmentation. The key idea is to identify concepts that have led to misclassifications and augment the dataset with images that contain these concepts. We design a utility function that computes the degree of match between the visual features in an image and the set of concepts in the under-represented regions. The proposed approach enables us to select samples with high utility scores and increase the representativeness of the training dataset. Experiment results on multiple datasets demonstrate that BRACE outperforms state-of-the-art augmentation methods.

## 2    Related Work

Data augmentation is a widely used pre-processing step in training deep neural networks. The objective is to inflate the training dataset with label preserving small transformations to avoid over-fitting. Techniques include geometric transformations, color space augmentations, kernel filters, mixing images, feature space augmentation, adversarial training, generative adversarial networks, neural style transfer, and meta-learning [3]. However, these augmentation methods only explore the neighborhood of existing images and are not able to add samples from under-represented regions [18].

The work in [6] taps into the large online image repositories to augment the training dataset. Object bounding boxes and part landmarks annotations in the original dataset are used to select relevant and diverse images. Since the images retrieved are typically noisy, [19] trains two models, to select in-distribution samples and to re-label the noisy images in the selected set. Both these works may not increase the representativeness of the dataset as they do not consider the under-represented regions in the dataset when deciding which samples to add.

Existing explanation methods for deep neural network based image classifiers include pixel-level attribution methods and high-level semantic explanation methods. Pixel-level attribution methods [8, 9, 10] assign an importance score to each pixel in the image based on the contribution of that pixel towards the model decision. On the other hand, semantic explanation methods explain model decisions in terms of human-friendly concepts and can be categorized into post-hoc methods versus fully interpretable classifiers. Post-hoc methods include Grad-CAM [11], FLEX [15], IBD [13], TCAV [20], and ACE [21]. The first three approaches are image-level explanation mechanisms explaining the model's prediction of the given input. In contrast, TCAV and ACE are class-level explanation mechanisms that explain the characteristics of the entire class. Fully interpretable classifiers such as ProtoPNet [16] and CCNN [17] explain model decisions in terms of prototypical parts and word phrases, respectively. The proposed BRACE framework is able to utilize explanations from both post-hoc methods and fully interpretable classifiers to obtain new samples for data augmentation.

## 3    Proposed Framework

Machine learning models often make wrong classifications for classes that have insufficient variance in training data. Adding more diverse samples to the mix will enable the model to improve its accuracy. Intuitively, we want to give more training samples to classes with higher misclassifications so that the model is able to learn the real-world variations.

Let $C$ be the set of class labels, and $D = \{(x_i, y_i)\}$ be the original training set, where $y_i \in C$ is the class label of image $x_i$. For each class $c \in C$, we want to augment $D$ with a set of new samples that is proportional to the number of misclassifications for that class. Given a deep neural network model

$M$, we compute the ratio of misclassifications $r_c$ for class $c$ on the validation set $D^v$ as follows:

$$r_c = 1 - \frac{\sum_{(x,c) \in D^v} \mathcal{I}(M(x) = c)}{m} \tag{1}$$

where $m$ is the number of validation samples with class label $c$, and $\mathcal{I}$ is a function that returns 1 when its argument is true and 0 otherwise. We will augment $D$ with $|D_c| \times r_c$ number of new samples where $D_c \subset D$ is the set of images with class label $c$.

Let $X_c$ be the set of samples obtained using class $c$ as the keyword query on image repositories. This set is typically noisy, which may degrade model performance. As such, we will assess whether the new image is in the under-represented region and whether it contains concepts that led to the misclassification. We say that $x$ is an under-represented sample if it has visual features of class $c$, but the model's confidence that $x$ belongs to a different class $\bar{c} \neq c$ is high. We compute the under-representation score as follows:

$$\beta(x, c, \bar{c}) = \frac{f_x \cdot f_c}{\|f_x\| \|f_c\|} \times e^{P(\bar{c}|x)} \tag{2}$$

where $f_x$ is the feature vector of $x$ and $f_c$ is the average feature vector of all the images in class $c$, $P(\bar{c}|x)$ is the predicted probability of $x$ belonging to class $\bar{c}$. Feature vectors are extracted from model $M$. A high $\beta$ value indicates that $x$ is similar to some images in class $c$, but the model has classified it as class $\bar{c}$. Such samples are more informative, and using them for training may help $M$ generalize better.

Further, the sample $x$ should not only possess features of class $c$ but should also contain specific concepts that caused the model $M$ to misclassify it as class $\bar{c}$. Adding images with these specific concepts will help the model to better differentiate class $c$ from class $\bar{c}$. Let $\mathcal{S}_{c \to \bar{c}}$ be the set of concepts that caused $M$ to misclassify images of class $c$ into $\bar{c}$, and $\bar{C} = \bigcup \bar{c}$. Let $\mathbf{\Delta}(\mathcal{S}_{c \to \bar{c}}, x)$ be the function that computes the degree of match between the visual features in an image $x$ and the concepts in $\mathcal{S}_{c \to \bar{c}}$. The derivation of $\mathcal{S}_{c \to \bar{c}}$ and $\mathbf{\Delta}(.)$ differ depending on the explanation mechanism used (see Section 4). The utility score of a sample $x$ is computed as follows:

$$utility(x) = \sum_{\bar{c} \in \bar{C}} [\beta(x, c, \bar{c}) \times \Delta(\mathcal{S}_{c \to \bar{c}}, x))] \tag{3}$$

Then $|D_c| \times r_c$ samples with the highest utility scores are added to the original train dataset. We repeat this process for all the classes in $D$. We fine-tune the weights of $M$ with the augmented training dataset. Note that adding new samples may result in data imbalance, and this is resolved by using weighted cross-entropy loss during fine-tuning where the weight of class $c$ is given by $\frac{1}{|D_c|(1+r_c)}$. With this, there will be minimal change to the weights of the classes where M already performs well. Algorithm 1 gives the details of the BRACE framework.

## 4 Derivation of $\mathcal{S}_{c \to \bar{c}}$ and $\Delta(.)$

In this section, we describe how we derive the concepts that cause the misclassification, $\mathcal{S}_{c \to \bar{c}}$, for two popular explanation mechanisms: the fully interpretable classifier with linguistic explanation and the black-box classifier with post-hoc explanation using heat maps. We also show how to compute the degree of match $\Delta(.)$ in these cases.

### 4.1 Fully Interpretable Classifier with Linguistic Explanation

The work in [17] introduces an additional concept layer to the CNN-based architecture to guide the learning of the associations between visual features and word phrases extracted from image descriptions. This Comprehensible Convolutional Neural Network (CCNN) explains its decisions in word phrases corresponding to different visual concepts. The training objective function considers *concept uniqueness* which encourages each learned concept to correspond to only one word phrase, and *mapping consistency* which aims to preserve the distance between the learned concept and its corresponding word phrase in a joint embedding space. Together with classification accuracy loss, CCNN is currently the state-of-the-art fully interpretable classifier. Here, we discuss how the proposed BRACE framework can further improve CCNN classification accuracy.

**Algorithm 1:** BRACE

**input** : Classification model $M$, Original training dataset $D$, Set of class labels $C$
**output** : Fine-tuned weights $\theta$

$\theta \leftarrow$ Weight initialization. $\qquad\qquad\qquad\qquad$ ▷ Initialize $\theta$ with pre-trained weights of $M$
**for** $iter \in [0, max\_iterations]$ **do**
 $D' \leftarrow D$
 **for** $c \in C$ **do**
  $r_c \leftarrow$ Calculate miss-classification ratio as in Eq. 1.
  $X_c \leftarrow$ Obtain images given the class name of $c$ as the query term.
  **for** $x \in X_c$ **do**
   **for** $\bar{c} \in \bar{C}$ **do**
    $\beta(x, c, \bar{c}) \leftarrow$ Calculate under-representation score as in Eq. 2.
    $\mathcal{S}_{c \rightarrow \bar{c}} \leftarrow$ Derive concepts caused misclassifications.
    $\Delta(\mathcal{S}_{c \rightarrow \bar{c}}, x) \leftarrow$ Calculate availability of concepts $\mathcal{S}_{c \rightarrow \bar{c}}$.
   **end**
   $utility(x) = \sum_{\bar{c} \in \bar{C}} [\beta(x, c, \bar{c}) \times \Delta(\mathcal{S}_{c \rightarrow \bar{c}}, x))]$
  **end**
  $D' \leftarrow D' \bigcup \{|D_c| \times r_c \text{ samples with the highest utility scores}\}$
 **end**
 $\theta \leftarrow$ finetune_weights($M$, $D'$)
**end**

Let $V = \{v_1, ..., v_n\}$ be the set of visual features extracted by the concept layer of CCNN for an image $x$. Each $v_i \in V$ corresponds to a concept described by a word phrase $p \in P = \{p_1, ..., p_n\}$. CCNN employs a Global Average Pooling (GAP) layer to express its classification decision as a weighted sum of the learned concepts. Suppose the output of the GAP layer is $\{o_1, ..., o_n\}$ where $o_i$ corresponds to concept $i$ in $x$. Then the contribution of $i$ to the class decision $c$ is given by $o_i \times w_{i,c} \in W$, where $W \in \mathcal{R}^{n \times C}$ is the weight matrix of the fully connected layer in CCNN. We apply the softmax function to obtain the percentage contributions and select the set of concepts whose contribution is greater than some threshold. We will take the top-5 if there are more than five concepts that satisfy the criteria. Here, we set the threshold to be $\frac{1}{n}$. For instance, CCNN classifies the bird in Figure 2 as Nashville Warbler and explains that the concepts "Yellow breast", "Grey crown" and "White eyering" accounted for 47%, 19%, and 14% of the decision, respectively.

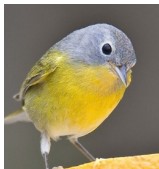

**Predicted class:** Nashville Warbler
**Concepts contributed:**
Yellow breast (0.47),
Grey crown (0.19) ,
White eyering (0.14)

Figure 2: Concepts contributed to CCNN's predictions.

Given $X_{c \rightarrow \bar{c}}$, the set of images with class label $c$ that have been misclassified as $\bar{c}$, we compute the number of images in $X_{c \rightarrow \bar{c}}$ that contains the concept $i$. The top $k$ concepts with the highest number form the set $\mathcal{S}_{c \rightarrow \bar{c}}$. We set $k = 3$ in our experiments. Recall, each visual feature $v_i$ in CCNN concept layer is mapped to a word phrase $p_i$ describing a concept, and $v_i$ is activated only when the concept occurs in the image. As such, given the set of concepts $\mathcal{S}_{c \rightarrow \bar{c}}$, we utilize their corresponding visual feature activations to determine the degree of match of a concept $i$ in an image $x$ as follows:

$$\alpha_i = -\log \left[ 1 - \frac{1}{1 + e^{-\bar{o}_i}} + \epsilon \right] \tag{4}$$

where $\bar{o}_i$ is the normalized $o_i$ obtained by subtracting the mean activation of the GAP layer. The negative log function gives more weight to an image with one highly activated concept than one with multiple weakly activated concepts. With this, the degree of match between the set $\mathcal{S}_{c \rightarrow \bar{c}}$ and image $x$ is given by:

$$\Delta(\mathcal{S}_{c \rightarrow \bar{c}}, x) = \sum_{i=1}^{k} \alpha_i \tag{5}$$

## 4.2 Post-hoc Explanation

BRACE requires image-level explanation to identify concepts that caused misclassification. One widely accepted image-level post-hoc explanation mechanism is Grad-CAM [11]. Grad-CAM computes gradients of the predicted class score with respect to the last convolutional layer feature maps. The gradients are averaged channel-wise and multiplied with the respective feature maps to create a coarse-grained saliency map. Since feature maps in the last convolutional layer correspond to semantic concepts [22, 23], we consider the image regions highlighted by GradCAM to cover semantic concepts. In our implementation, we normalize the pixels in the saliency map to have values between 0 and 1, and regions whose values are greater than the threshold of 0.5 correspond to semantic concepts that are responsible for the model decision.

Let $\mathcal{X}_{c \to \bar{c}}$ be the set of images whose class label has been misclassified from $c$ to $\bar{c}$. Then $\mathcal{S}_{c \to \bar{c}}$ is the set of regions in $\mathcal{X}_{c \to \bar{c}}$ obtained from the saliency maps of Grad-CAM. The regions in $\mathcal{S}_{c \to \bar{c}}$ are passed through $M$, and the output of the last layer before the classification layer form their visual features, denoted as $U$. Given an image $x$, we use a state-of-the-art RCNN model [24] to derive salient region proposals that correspond to an object or object part. These region proposals are passed through $M$ to obtain the corresponding visual features denoted as $W$. The degree of match between $\mathcal{S}_{c \to \bar{c}}$, and $x$ is given by the similarity between visual features in $U$ and $W$.

$$\Delta(\mathcal{S}_{c \to \bar{c}}, x) = \sum_{u \in U} z_u \quad \text{where} \ \ z_u = \max_{w \in W} \left( -\log \left[ 1 - \frac{u.w}{\|u\|\|w\|} + \epsilon \right] \right) \tag{6}$$

An image with high $\Delta(\mathcal{S}_{c \to \bar{c}}, x)$ indicates that it contains many concepts that led the model to misclassify. Such images help the model to learn correct decision boundaries and are selected to augment the training dataset. For each concept $u$ that causes misclassifications, $z_u$ indicates the degree of match between $u$ and some region $w$ in $x$. We take the region with the maximum similarity as it indicates the highest probability of $u$ being present in the image. Using a negative log allows us to choose an image containing at least one high confidence concept that causes misclassification over another image with multiple low confidence concepts. This approach can be utilized when we use BRACE with any explanation method that provides super-pixel-based explanations, e.g., ACE [21] and IBD [13].

## 5  Performance Evaluation

We carry out experiments to show that BRACE can effectively improve the accuracy of general image classification tasks as well as fine-grained image classification. We also demonstrate the broad applicability of BRACE on a wide range of network architectures such as ResNet[25] and DenseNet[26]. All the codes are implemented using PyTorch, and the experiments are run on NVIDIA Tesla V100 GPUs. The following datasets are used:

1. Caltech UCSD Birds (CUB) [27]. This dataset has 11,788 images of birds belonging to 200 classes. The dataset is divided into a train set of 3994 images, a validation set of 2000 images, and a test set of 5794 images. Each image has ten sentences describing the bird collected by [28].

2. CUB-Families [29]. This dataset groups the 200 species of birds in CUB into 37 families comprised of multiple bird species. We create under-represented regions by removing some species in the test split of [30] from each family in the training dataset. No species are removed in the validation and test sets. The resultant dataset contains 4585 training images, 2343 validation images, and 4860 test images.

3. Tiny ImageNet. Tiny ImageNet contains 110,000 images with 200 classes. In the standard dataset split, each class has 500 training images (400 for training the model and 100 for tuning the hyper-parameters) and 50 validation images. We report our results on the validation set.

In our experiments, we use the samples collected by [31] for CUB and CUB-Families. Since [31] has not collected samples for Tiny ImageNet, we use Flicker API to obtain 250 new samples per class using the class name as the search query. These samples are used by all the methods in our comparative study.

The number of samples added to each class $c$ is based on the ratio of misclassifications $r_c$ (recall Section 3). To ensure that no two images are highly similar, we remove images similar to the ones in the training image set and duplicates obtained from the online repositories.

Since CCNN requires an indicator vector per image for training, we extract 398 and 378 word phrases from the image descriptions of CUB and CUB-Families, respectively, to construct these vectors depicting the concepts present in each image. The union of the indicator vectors in the same class are called the *class indicator vectors*. For each dataset, we set the number of nodes in the concept layer to be the same as the number of extracted word phrases and train CCNN using the same hyper-parameters settings in [17]. For new samples without image descriptions, we use the corresponding class indicator vector as the indicator vectors.

## 5.1 Comparison with Data Augmentation Methods

We compare BRACE and the following state-of-the-art data augmentation methods:
- **CutMix** [5]. This mixed-based data augmentation method cuts out one image patch, pastes it on another image, and mixes their labels according to the area proportion.
- **SnapMix** [4]. This is another mixed-based data augmentation method. It combines images similar to CutMix but uses the semantic composition of the resultant image to derive the label.
- **WS-DAN** [32]. WS-DAN is an attention-based approach that uses attention-cropping and attention-dropping. Attention-cropping refers to cropping the region of the image attended by the model to create a new sample. Attention-dropping refers to erasing the attended region to create a new sample.
- **Part-based** [6]. This method obtains samples from image repositories by using object bounding boxes and part landmark annotations.
- **Metaset-based** [19]. This method also obtains samples from image repositories. It trains two models to ensure the samples are in-distribution and to correct noisy labels.

We train a black-box classifier using ResNet-34 architecture. Table 1 shows the results on CUB, CUB-Families, and Tiny ImageNet. We observe that training using the BRACE-augmented datasets consistently achieves the highest accuracy for all datasets. In contrast, Cut-mix and Snap-mix fall short as they only combine existing images without considering the under-represented regions. Though WS-DAN uses the attention weights to determine the regions for cropping/dropping, the samples created may not cover the under-represented regions, leading to a lower accuracy compared to BRACE. For the CUB dataset, Part-based augmentation added a total of 106,000 samples while Metaset-based added 18,000 samples. Despite the large number of new samples, the improvements are lower compared BRACE, indicating the importance of adding only samples that are likely to cover the under-represented regions.

Table 1: Performance of ResNet-34 black-box classifier with GradCAM.

| Method | CUB | CUB-Families | Tiny ImageNet |
|---|---|---|---|
| Original dataset | 85.5 | 82.6 | 76.1 |
| Cut-mix | 85.7 | 85.9 | 77.0 |
| Snap-mix | 87.1 | 86.3 | 77.5 |
| WS-DAN | 87.2 | 83.5 | 76.6 |
| Part-based | 86.3* | - | - |
| Metaset-based | 86.8 | 89.3 | 73.1 |
| BRACE | **87.7** | **90.0** | **78.3** |

*taken from corresponding references.

Next, we examine the performance of a fully interpretable classifier using these augmentation techniques. We train a CCNN based on the ResNet-34 architecture. Note that when mixed-based augmentations are used with CCNN, the indicator vector of the original image might not match the resultant image. Hence, we combine the class indicator vectors of mixed classes according to their proportions and apply a threshold to create the indicator vector of the resultant image. Further, CCNN requires image descriptions that are available in the CUB and CUB-Families datasets. We omit the Tiny ImageNet dataset as it does not have image descriptions. Table 2 shows the results on CUB and

Table 2: Performance of fully interpretable CCNN classifier based on ResNet-34.

| Method | CUB | CUB-Families |
|---|---|---|
| Original dataset | 84.3 | 83.8 |
| Cut-mix | 80.6 | 79.0 |
| Snap-mix | 82.4 | 79.9 |
| WS-DAN | 81.6 | 81.8 |
| Metaset-based | 85.1 | 88.1 |
| BRACE | **86.1** | **88.7** |

CUB-Families. Again, we see that BRACE-augmented datasets achieve the highest accuracy. The improvement is bigger in CUB-Families, where there are more under-represented regions.

## 5.2 Comparison of Sample Selection Methods

BRACE ranks the samples based on the utility scores and selects those with top scores to augment the training dataset. In this set of experiments, we compare BRACE with the following methods:

- **Random.** Here, the samples are selected randomly.
- **Confidence.** The samples obtained for each class are passed through the classifier. We rank the samples based on the classifier's scores, and the top scored samples are selected.
- **Core-set [33]**. k representative samples are selected using the K-Center-Greedy algorithm.
- **L-loss [34]**. The classification loss of a sample is predicted, and samples with the top-k losses are returned.

Table 3 shows the classification accuracies for ResNet-101 and DenseNet-161 based classifiers using post-hoc explanation. We see that Brace(utility) has the greatest improvement in accuracy in all datasets. The results for L-loss and Core-set are mixed when compared to Random and Confidence methods. This is because both L-loss and Core-set tend to select more noisy images (e.g., out-of-distribution images) as compared to Random and Confidence leading to lower performance accuracy. See Figure 4.

Table 3: Comparison of classification accuracies using post-hoc explanation.

| Method | CUB | | CUB-Families | | Tiny ImageNet | |
|---|---|---|---|---|---|---|
| | Dense-161 | Res-101 | Dense-161 | Res-101 | Dense-161 | Res-101 |
| Original dataset | 86.6 | 87.4 | 86.5 | 85.4 | 80.1 | 81.3 |
| Core-set | 84.8 | 85.9 | 87.2 | 88.6 | 80.2 | 77.8 |
| L-loss | 85.2 | 84.8 | 86.5 | 88.4 | 78.3 | 77.4 |
| Random | 86.5 | 86.4 | 87.0 | 88.3 | 80.3 | 76.8 |
| Confidence | 87.3 | 87.0 | 86.5 | 86.6 | 80.6 | 81.3 |
| BRACE(utility) | **88.0** | **89.2** | **93.0** | **91.2** | **81.1** | **81.7** |

Table 4: Comparison of classification accuracies using linguistic explanation mechanism.

| Method | CUB | | CUB-Families | |
|---|---|---|---|---|
| | Dense-161 | Res-101 | Dense-161 | Res-101 |
| Original dataset | 84.5 | 86.6 | 85.8 | 85.7 |
| Core-set | 85.0 | 84.5 | 87.1 | 86.1 |
| L-loss | 82.0 | 84.7 | 84.7 | 85.5 |
| Random | 85.8 | 87.0 | 88.6 | 88.0 |
| Confidence | 85.4 | 86.7 | 85.8 | 85.8 |
| BRACE(utility) | **86.8** | **88.4** | **91.9** | **92.2** |

Table 4 shows the classification accuracy when we use the linguistic explanation mechanism. We observe that the proposed BRACE(utility) consistently gives the highest accuracy. We also investigate the performance of BRACE(utility) in the under-represented classes by selecting five classes in the CUB-Families dataset that have the lowest classification accuracy. Table 5 shows that BRACE(utility) significantly improves the classification accuracies of these classes compared to other methods. Note that Confidence has a lower accuracy compared to Random. One possible reason is that the samples selected by Confidence tend to be similar to the training images resulting in model over-fitting.

Table 5: Class-level classification accuracies on different categories in CUB-Families.

| Method | Hirundinidae | Mimidae | Fringillidae | Cuculidae | Icteridae |
|---|---|---|---|---|---|
| Original dataset | 41.1 | 46.7 | 52.3 | 53.1 | 55.4 |
| Core-set | 54.7 | 67.6 | 72.7 | 56.3 | 67.7 |
| L-loss | 52.8 | 66.7 | 68.2 | 49.2 | 72.1 |
| Random | 67.1 | 70.2 | 63.9 | 63.3 | 70.8 |
| Confidence | 32.9 | 44.4 | 56.9 | 53.1 | 50.0 |
| BRACE(utility) | **72.6** | **71.1** | **75.6** | **67.3** | **82.2** |

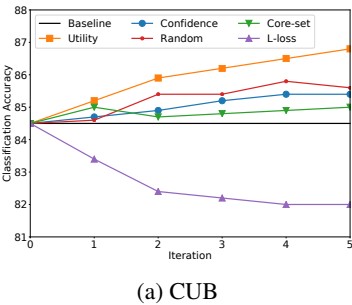

(a) CUB

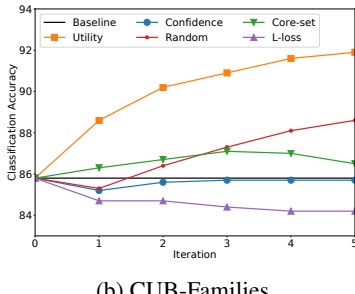

(b) CUB-Families

Figure 3: Classification accuracy of DenseNet-based CCNN.

Figure 4: Comparison of samples selected by different methods.

Figure 3 shows the classification accuracy of the DenseNet-161-based CCNN for the different iterations as we add samples to the training dataset. We see that with just one iteration, BRACE(Utility) is able to achieve higher accuracies compared to the other methods. This indicates that the utility score can identify the under-represented, and hence more informative samples are added to the training dataset, thus enabling the learning of such under-represented class boundaries. This performance gap widens in the CUB-Families dataset.

Finally, we visualize the samples selected for the "Hirundinidae" and "Fringillidae" categories in CUB-Families. Figure 4 shows that the samples selected by BRACE(utility) are the most similar to the images in the subcategories that have been removed. This assures us that BRACE(utility) can select samples containing concepts of the under-represented regions. This allows the model to learn the correct boundaries for these under-represented regions, leading to higher accuracies.

## 5.3 Ablation Study

Next, we examine the effects of $\beta$ and $\Delta$ by implementing two variants of BRACE with CCNN:

- BRACE$^{-\Delta}$ does not take into account the degree of match with the concepts that caused misclassification. In other words, the utility score is given by $\sum_{\bar{c}\in\bar{C}}\beta(x,c,\bar{c})$
- BRACE$^{-\beta}$ does not consider if a sample falls in the under-represented region and computes the utility score as $\sum_{\bar{c}\in\bar{C}}\Delta(\mathcal{S}_{c\to\bar{c}},x)$.

Table 6 shows the classification accuracy on CUB and CUB-Families datasets. We observe that the highest accuracy is achieved when both $\beta$ and $\Delta$ are used in the computation of utility score. This is because $\beta$ ensures that the samples selected fall into the under-represented regions, while $\Delta$ ensures that the concepts that caused misclassifications are present in the selected samples. Further, we see that $\Delta$ plays a greater role in improving the classification accuracy compared to $\beta$.

Table 6: Results of ablation study.

| Method | CUB | | CUB-Families | |
|---|---|---|---|---|
| | Dense-161 | Res-101 | Dense-161 | Res-101 |
| CCNN | 84.8 | 86.6 | 85.8 | 85.7 |
| BRACE$^{-\Delta}$ | 85.9 | 87.1 | 91.2 | 91.4 |
| BRACE$^{-\beta}$ | 86.3 | 87.6 | 91.3 | 91.6 |
| BRACE | **86.8** | **88.4** | **91.9** | **92.2** |

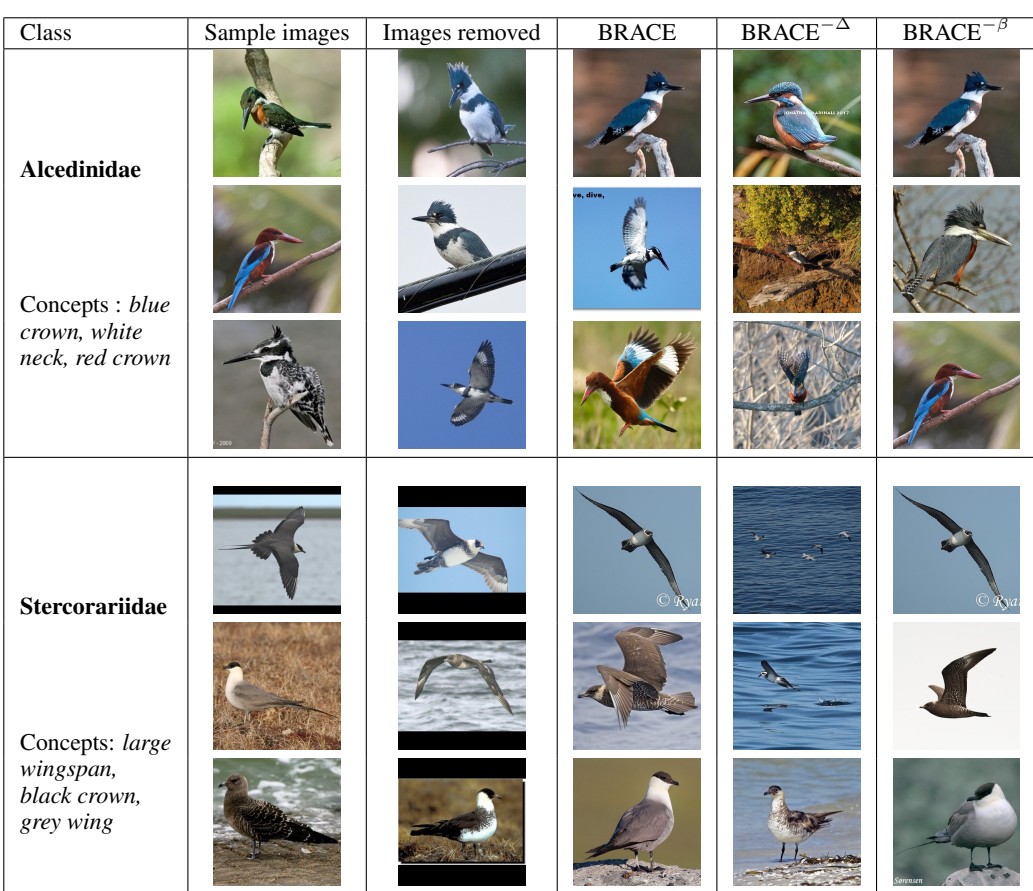

Figure 5: Comparison of samples selected by variants of BRACE.

Figure5 shows samples selected by BRACE variants for two classes in CUB-Families given the top-3 concepts responsible for the misclassifications for each class. We observe that BRACE$^{-\Delta}$ has selected images from under-represented regions. They either belong to the subcategories that were

excluded from the training dataset, have unfamiliar backgrounds, or the discriminative features of the class are hidden due to the pose or occlusions. However, the images selected by BRACE$^{-\Delta}$ do not include the concepts caused for misclassifications. On the other hand, BRACE$^{-\beta}$ selects images clearly showing one or more concepts that have led to misclassifications, e.g., Alcedinidaes with a white neck and Stercorariidae with a large wingspan. However, some of these images do not fall into under-represented regions. In contrast, BRACE has selected images from the under-represented regions that contain concepts contributed for misclassifications. Such images are informative and help the model to learn correct boundaries, leading to higher accuracies.

We compute the "overlapping rate", defined as the intersection over the union of sets of the samples selected by BRACE variants. The overlapping rate of BRACE$^{-\Delta}$ and BRACE$^{-\beta}$ is 0.09, indicating that the two variants tend to select different image sets. In addition, the overlapping rate between BRACE and BRACE$^{-\Delta}$ is 0.20, which is lower than the 0.37 overlapping rate between BRACE and BRACE$^{-\beta}$. This suggests that $\Delta$ has a greater influence on the set of images selected by BRACE.

## 5.4 Generalizability with BRACE

A representative dataset facilitates the learning of more generalizable features. As a result, a classifier trained on such a representative dataset can recognize a wide variety of instances of a class even when some of those variations were not part of the training data for that class. We demonstrate the generalizability of BRACE in this final set of experiments.

We use ResNet-34 trained on CUB-Families and test on a subset of NAbirds [35]. This subset consists of classes that are found in CUB-Families. These classes have a wide variety of sub-categories that may or may not be present in the CUB-Families. We call this subset NAbirds-Sub. For ResNet-34 trained on Tiny ImageNet, we test it on the subset of images from ImageNet-V2 [36]. This subset consists of classes that are in Tiny ImageNet, and we call it ImageNet-V2-Sub. Table 7 shows that BRACE(utility) achieves the highest accuracy, confirming that BRACE enables the classifier to learn features that are generalizable to handle more diverse images. Note that when collecting new images for Tiny ImageNet using Ficklr API, we set the API parameters different from those used in [36] to ensure there is no overlap between the images we collect and ImageNet-V2.

Table 7: Comparison on generalizability.

| Method | NAbirds-Sub | ImageNet-V2-Sub |
|---|---|---|
| Original dataset | 81.5 | 54.4 |
| Cut-mix | 72.1 | 37.2 |
| Snap-mix | 76.0 | 42.6 |
| WS-DAN | 67.0 | 56.1 |
| Metaset-based | 83.5 | 42.4 |
| Core-set | 83.0 | 61.9 |
| L-loss | 79.7 | 63.9 |
| Random | 82.4 | 65.1 |
| Confidence | 81.5 | 67.6 |
| BRACE(utility) | **84.9** | **70.0** |

## 6 Conclusion

In this work, we have examined the potential of utilizing explanation mechanisms to augment training datasets to increase classification accuracy. In contrast to existing data augmentation approaches, we have proposed a framework called BRACE that uses the explanations from model decisions to understand the limitations in the training dataset and select informative images that are in the under-represented regions and contain visual concepts that have led to misclassifications to fine-tune the classifier. The assumption here is that the image repositories contain a sufficiently large number of diverse images. For applications with limited images (e.g., images of rare diseases), the performance of BRACE may be affected. Nevertheless, the experiment results on multiple datasets demonstrate that our proposed solution is able to learn features that are generalizable and improve the classification accuracy over state-of-the-art augmentation methods.

**Acknowledgement.** This research is supported by the National Research Foundation, Singapore under its AI Singapore Programme (Award No: AISG-GC-2019-001, AISG-RP-2018-008).

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
