# Explanation-based Data Augmentation for Image Classification - Supplementary

**Sandareka Wickramanayake**     **Mong Li Lee**     **Wynne Hsu**
School of Computing
National University of Singapore
{sandaw, leeml, whsu}@comp.nus.edu.sg

## 1   Links and Licences of Datasets Used

All the datasets used in our paper are publicly available and are to be used for research purposes. To the best of our knowledge, they do not have any personally identifiable information or offensive content. Table 1 gives the download links and licenses of these datasets.

Table 1: Download links and licences of Datasets.

| Dataset | Download Link and License |
| --- | --- |
| Caltech UCSD Birds (CUB) (1) | `http://www.vision.caltech.edu/visipedia/CUB-200-2011.html` |
| | Use is restricted to non-commercial research and educational purposes |
| CUB-Families (2) | `https://github.com/HCPLab-SYSU/HS` |
| | Use is restricted to non-commercial research and educational purposes |
| Tiny ImageNet | `http://cs231n.stanford.edu/tiny-imagenet-200.zip` |
| | Use is restricted to non-commercial research and educational purposes |

We use the samples collected in (3) as the image repository for CUB and CUB-Families. The image set collected in (3) can be downloaded at `https://wsnfg-sh.oss-cn-shanghai.aliyuncs.com/web-bird.tar.gz`. This image set is noisy and includes out-of-distribution samples. Sample images are shown in Figure1.

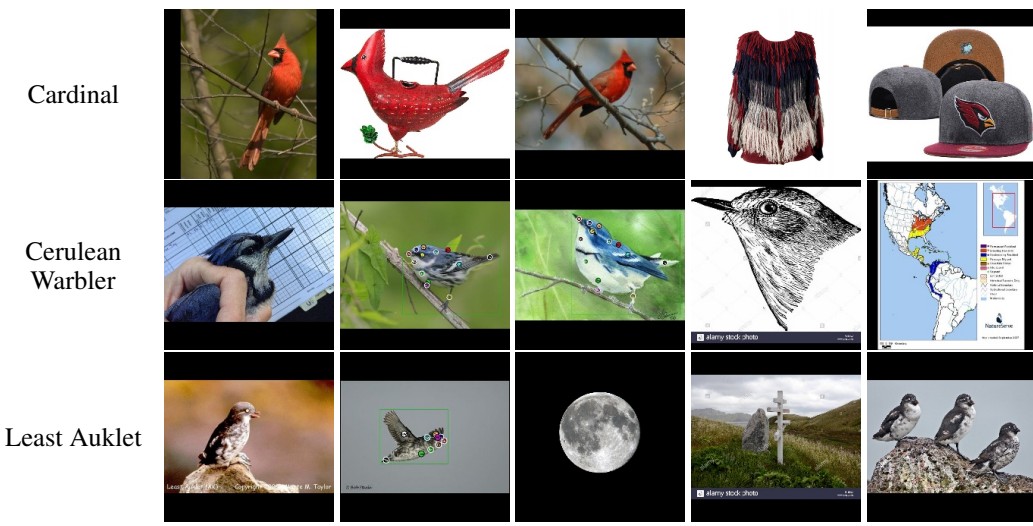

Figure 1: Samples images for three classes of CUB dataset collected in (3).

35th Conference on Neural Information Processing Systems (NeurIPS 2021).

Since (3) does not collect samples for Tiny ImageNet, we use the Flickr image hosting service via the Flicker API to obtain 250 new samples per class using the class name as the search query. Note that when collecting new images for Tiny ImageNet using Ficklr API, we set the API parameters different from those used in (4) to ensure there is no overlap between the images we collect and ImageNet-V2. The collected image set can be downloaded at `https://bit.ly/3pkd4BE`. Sample images are shown in Figure2.

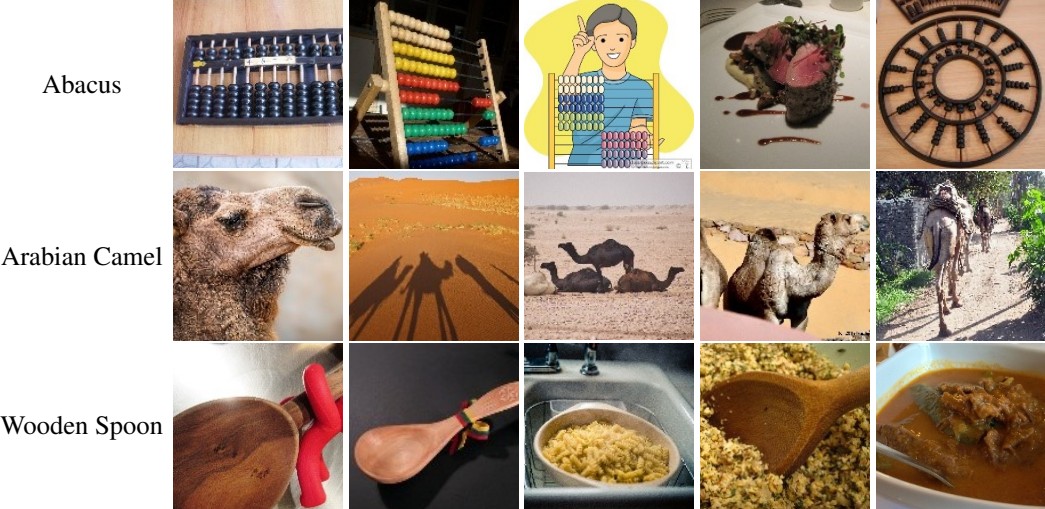

Figure 2: Samples images for three classes of Tiny-ImageNet dataset collected using Flickr API.

These newly collected sample sets sometimes include duplicate samples or samples very similar to the ones in the original training dataset. Adding such samples does not contribute to improve the classification accuracy. Hence, before selecting the samples to be added, we remove the duplicates or samples very similar to existing training data. We consider two images are similar if the cosine similarity between them is greater than 0.99.

## 2 Implementation Details

All the codes are implemented using PyTorch, and the experiments are run on NVIDIA Tesla V100 GPUs. We use publicly available source code of Snap-mix (5), WS-DAN (6), and Metaset-based (7) to obtain results for the respective methods. For Cut-mix, we use the source code provided in the SnapMix repository. We use code provided by (8) for the experiments of Core-set and L-loss methods.

We utilize ResNet-34, ResNet-101 (9) and DenseNet(10) pre-trained on ImageNet(11) and fine-tune them using the standard SGD (12) with the momentum of 0.9, weight decay of $5e^{-}4$. We set the initial learning rate to $1e^{-}4$ for the training with original data and $1e^{-}5$ for the training after augmenting the dataset with BRACE. Subsequently, the learning rate is decayed with a cosine annealing (13). We train the models for 50 epochs. The batch size is set as 16 for CUB and CUB-Families and 72 for Tiny-ImageNet. The same setting is used for training models on the datasets augmented with Random and Confidence selection methods. For other data augmentation methods, the default hyper-parameters given in the official source code were used. The meta dataset in Metaset-based (7) was set to the original training dataset.

The number of samples added to each class $c$ is based on the ratio of misclassifications $r_c$. We set $max\_iterations = 5$. Since CCNN requires an indicator vector per image for training, we extract 398 and 378 word phrases from the image descriptions of CUB and CUB-Families, respectively, to construct these vectors depicting the concepts present in each image. The union of the indicator vectors in the same class are called the *class indicator vectors*. For each dataset, we set the number of nodes in the concept layer to be the same as the number of extracted word phrases and train CCNN using the same hyper-parameters settings in (14): select the top 20 phrases based on tf-idf scores as concepts for each class and set $\lambda = 0.4, \alpha = 1, \beta = 0.5$ and the joint embedding space to 24.

For new samples without image descriptions, we use the corresponding class indicator vector as the indicator vectors.

The number of images added by each method for each dataset is shown in Table 2.

Table 2: Number of images added by each method for different datasets.

| Method | CUB | CUB-Families | Tiny ImageNet |
|---|---|---|---|
| Metaset-based | 18000 | 18000 | 30000 |
| Random | 7000 | 2000 | 13000 |
| Confidence | 7000 | 2000 | 13000 |
| Core-set | 7000 | 2000 | 13000 |
| LLoss | 7000 | 2000 | 13000 |
| BRACE | 7000 | 2000 | 13000 |

## 3 Mean and Standard Deviation of Experiment Results

Table 3 shows the mean and standard deviation of classification accuracy for the black-box classifier using ResNet-34 architecture over three runs on CUB, CUB-Families, and Tiny ImageNet.

Table 3: Performance of ResNet-34 black-box classifier with GradCAM. Mean and standard deviation over three runs.

| Method | CUB | CUB-Families | Tiny ImageNet |
|---|---|---|---|
| Original Training Dataset | 85.5±0.04 | 82.6±0.24 | 76.1±0.08 |
| Cut-mix | 85.7±0.30 | 85.9±0.45 | 77.0±0.00 |
| Snap-mix | 87.1±0.30 | 86.3±0.31 | 77.5±0.60 |
| WS-DAN | 87.2±0.06 | 83.5±0.61 | 76.6±0.64 |
| Metaset-based | 86.8±0.32 | 89.3±0.20 | 73.1±0.16 |
| BRACE | **87.7±0.20** | **90.0±0.40** | **78.3±0.00** |

Table 4 shows the mean and standard deviation of classification accuracy for the fully interpretable CCNN on the ResNet-34 architecture over three runs on CUB and CUB-Families.

Table 4: Performance of fully interpretable CCNN classifier based on ResNet-34. Mean and standard deviation over three trials.

| Method | CUB | CUB-Families |
|---|---|---|
| Original Training Dataset | 84.3±0.14 | 83.8±0.13 |
| Cut-mix | 80.6±0.72 | 79.0±1.23 |
| Snap-mix | 82.4±0.34 | 79.9±1.25 |
| BRACE | **86.1±0.23** | **88.7±0.06** |

Table 5 shows the the mean and standard deviation of classification accuracy of the variants of BRACE with CCNN over three runs on CUB and CUB-Families datasets.

Table 5: Results of ablation study. Mean and standard deviation over three runs.

| Method | CUB | | CUB-Families | |
|---|---|---|---|---|
| | Dense161 | Res101 | Dense161 | Res101 |
| CCNN | 84.8±0.33 | 86.6±0.09 | 85.8±0.13 | 85.7±0.09 |
| BRACE$^{-\Delta}$ | 85.9±0.25 | 87.1±0.31 | 91.2±0.17 | 91.4±0.19 |
| BRACE$^{-\beta}$ | 86.3±0.41 | 87.6±0.49 | 91.3±0.26 | 91.6±0.10 |
| BRACE | **86.8±0.49** | **88.4±0.08** | **91.9±0.20** | **92.2±0.05** |

## 4 Paired T-test on the Experimental results

We conducted a paired t-test on classification accuracy achieved by BRACE and each baseline method for different datasets when post-hoc explanations are used and when explanations from CCNN are

used. The null hypothesis is that BRACE and a given baseline method have the identical average classification accuracy for the given dataset. Results are shown in Table6 and Table7. We observe that p-value is consistently below 0.05 except for one case (comparison with WS-DAN on CUB) indicating that accuracy improvement achieved by BRACE is statistically significant.

Table 6: P-values of the paired t-test on the results when post-hoc explanations are used.

| Method | CUB | CUB-Families | Tiny ImageNet |
|---|---|---|---|
| Original Training Dataset | 0.008 | 0.002 | 0.0004 |
| Cut-mix | 0.015 | 0.008 | 0.0007 |
| Snap-mix | 0.043 | 0.001 | 0.041 |
| WS-DAN | 0.079 | 0.002 | 0.039 |
| Metaset-based | 0.044 | 0.027 | 0.0007 |

Table 7: P-values of the paired t-test on the results explanations from CCNN are used.

| Method | CUB | CUB-Families |
|---|---|---|
| Original Training Dataset | 0.0007 | 0.001 |
| Cut-mix | 0.006 | 0.0005 |
| Snap-mix | 0.004 | 0.002 |

# 5 Classification Accuracy Comparison with State-of-the-art Methods on CUB Dataset

In this section, we compare performance of BRACE and state-of-the-art methods on CUB dataset. For a fair comparison we consider the state-of-the-art methods that use ResNet-101 as their backbone. Table 8 shows the performance comparison for black-box classifiers and Table 9 shows the comparison for fully interpretable classifiers. We observe that BRACE outperforms the existing techniques in both black-box as well as fully interpretable classification scenarios.

Table 8: Comparison of classification accuracy with the state-of-the-art black-box classifiers on CUB dataset.

| Model | Accuracy |
|---|---|
| MAMC (15) | 86.5 |
| DBT-Net (16) | 88.1 |
| Cut-mix (17) | 87.9 |
| API-Net (18) | 88.6 |
| Snap-mix (5) | 88.7 |
| BRACE | **89.2** |

Table 9: Comparison of classification accuracy with the state-of-the-art fully interpretable classifiers on CUB dataset.

| Model | Accuracy |
|---|---|
| CAM (19) | 85.7 |
| ProtoPNet (20) | 72.6 |
| CI-GC (21) | 77.6 |
| CCNN (14) | 86.6 |
| BRACE | **88.4** |

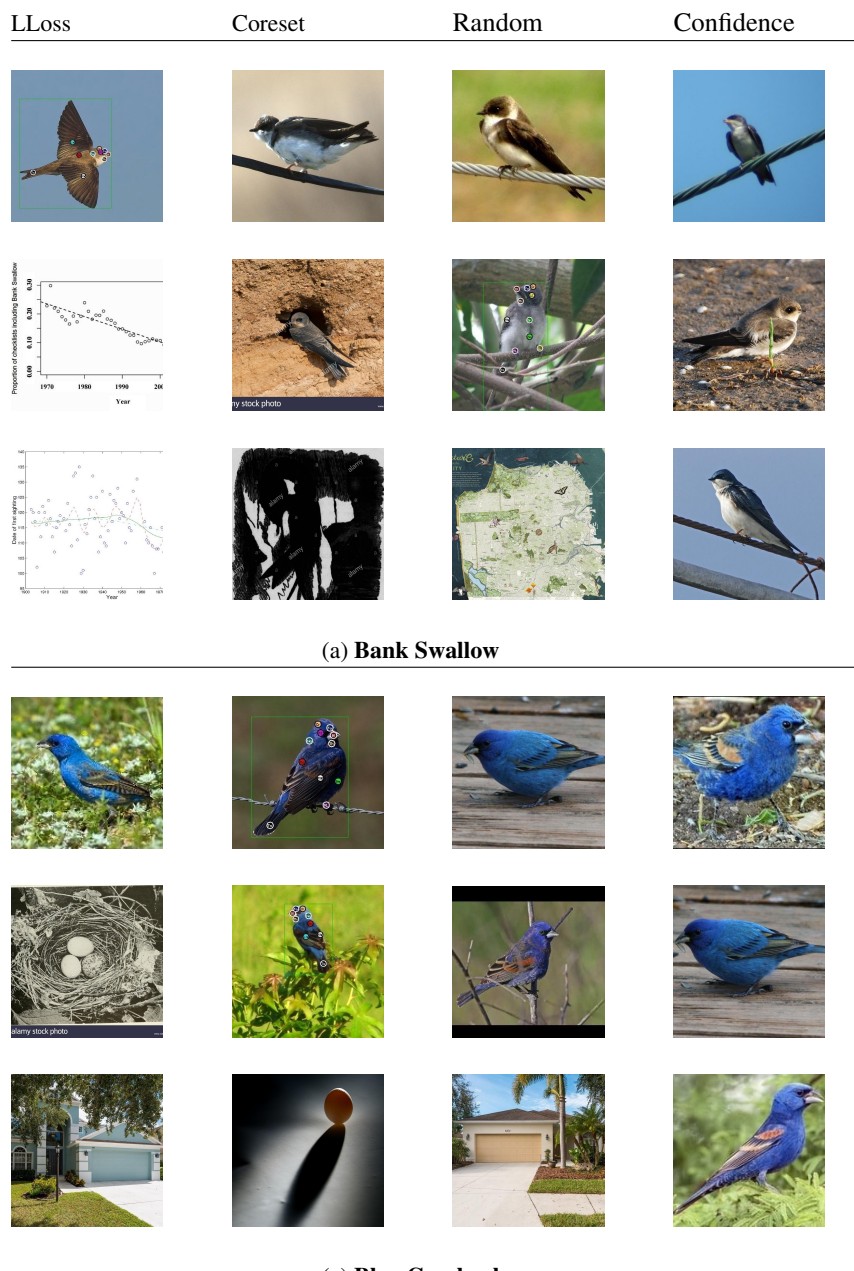

(a) **Bank Swallow**

(a) **Blue Grosbeak**

Figure 3: Comparison of samples selected by L-loss, Core-set, Random and Confidence methods.

Table 10: Comparison of classification accuracies using post-hoc explanation.

| Method | CUB | | CUB-Families | | Tiny ImageNet | |
|---|---|---|---|---|---|---|
| | Dense-161 | Res-101 | Dense-161 | Res-101 | Dense-161 | Res-101 |
| Original dataset | 86.6 | 87.4 | 86.5 | 85.4 | 80.1 | 81.3 |
| Core-set | 84.8 | 85.9 | 87.2 | 88.6 | 80.2 | 77.8 |
| L-loss | 85.2 | 84.8 | 86.5 | 88.4 | 78.3 | 77.4 |
| Random | 86.5 | 86.4 | 87.0 | 88.3 | 80.3 | 76.8 |
| Confidence | 87.3 | 87.0 | 86.5 | 86.6 | 80.6 | 81.3 |
| BRACE(utility) | **88.0** | **89.2** | **93.0** | **91.2** | **81.1** | **81.7** |

# 6 Comparison of Selection Methods

Table 10 shows the classification accuracies for ResNet-101 and DenseNet-161 based classifiers using post-hoc explanation. We see that Brace(utility) has the greatest improvement in accuracy in all datasets. The results for L-loss and Core-set are mixed when compared to Random and Confidence methods. This is because both L-loss and Core-set tend to select more noisy images (e.g., out-of-distribution images) than Random and Confidence leading to lower performance accuracy. We calculate the percentage of noisy images selected by each method in the top 1000 images. The results are shown in Table 11.

Table 11: Percentage of noisy images in the top 1000 images selected by different selection methods for CUB dataset.

| Method | Percentage of noisy images |
|---|---|
| Random | 17.6 |
| Confidence | 0.0 |
| Core-set | 21.0 |
| L-loss | 88.5 |

We see that both L-loss and Core-set tend to select more noisy images as compared to Random and Confidence leading to lower performance accuracy. Sample images selected by each method are given in Figure 3

In Figure 4, we show samples selected by BRACE(utility), Random, and Confidence selection methods for two more categories of the CUB-Families dataset, namely "Cuculidae" and "Icteridae". We observe that BRACE(utility) has selected the samples most similar to the images in the subcategories that have been removed. This assures us that BRACE(utility) can select samples containing concepts of the under-represented regions. In contrast, Confidence has selected samples similar to those are already in the training dataset, and Random has even chosen out-of-distribution samples.

# 7 Societal Impact

In this work, we propose BRACE, which utilizes explanation mechanisms to augment training datasets to increase classification accuracy. Experiment results indicate that BRACE improves the generalizability of classifiers and enables them to correctly classify more diverse images. Hence, BRACE will contribute to increasing the adaption of AI systems in the real world. However, the increased adaption of such systems might make some jobs obsolete. This can be identified as a potential indirect negative social impact of this work.

| Category | Example current train images | Example removed images | Samples selected by method | | |
|---|---|---|---|---|---|
| | | | Utility | Random | Confidence |
| Cuculidae | | | | | |
| Icteridae | | | | | |

Figure 4: Comparison of samples selected by different methods.