# OpenReview forum: "Explanation-based Data Augmentation for Image Classification"
_NeurIPS.cc/2021/Conference — NeurIPS 2021 Poster_

### Official Review · Reviewer_4Jog · 2021-06-25

**Rating:** 6
**Confidence:** 3

**Summary:**

This paper proposes a form of data augmentation that uses explanations to select external images from poorly performing classes.  Specifically, it looks for images that:
- are "under-represented" meaning they are similar (according to the model) to images of the target class but are predicted as being part of a different class
-  have the same concepts that are often present when the model mistakes images of the target class as being part of this different class

**Limitations And Societal Impact:**

Yes

**Main Review:**


# Originality

The methodology is novel and the idea of using explanations to guide data augmentation is an interesting one.  The connection between this work and prior work is clear.


# Quality

Most of the claims are well-supported, but there are a few things that need clarification.

*L128:  "We apply a softmax function to obtain the percentage contributions and select the set of concepts whose contribution is greater than some threshold."*

How is this threshold chosen?


*L150: "We consider the image patches whose pixels have importance scores greater than some threshold as the concepts that have contributed to the model decision."*

How is this threshold chosen?


*L175: "We report our results on the validation set."*

Based on the "Implementation Details" in the Supplement, it seems like the hyper-parameters for this method were tuned while those from the baselines were not.  How were the hyper-parameters tuned?  If the validation set was used, this places the results on this dataset into doubt.  This is especially concerning because CUB(-Family) is a small dataset, where we would expect augmentation to be more helpful than on larger datasets, and TinyImageNet is the only larger dataset tested.


*Table 2.*

Why are the results for the other "external source" data augmentation methods not shown?


*5.2 Comparison of Sample Selection Methods*

Why are these results based on ResNet101/DenseNet161 while the previous sections' results are based on ResNet34?


# Clarity

Most of the paper is clearly written.  Detailed comments follow

*2.  Related Work*
-  It would be nice if this section was clearly divided into "Augmentation" and "Explanation" based sections
-  It would be nice if relationship between [6] and [19] was clarified
-  It is a little unusual to describe methods like GradCAM as concept-based because they just point to parts of the image when methods like TCAV/ACE/CCNN are directly based on concepts.   Relatedly, it is surprising that only GradCAM and CCNN are used in Section 4.

*3.  Proposed Framework*
-  "A high β value indicates that x is similar to some image in class c but the model M has classified it as class c ̄. " This doesn't seem quite right because Equation 2 is based on similarity with the mean of the class representation rather than the max of the similarity images of the class.
-  Generally, a simple illustration would help explain the intuition behind the beta term (which otherwise requires some effort to realize why it is a "under-represented region")

*4.1 Fully Interpretable Classifier with Linguistic Explanation*
-  This section relies heavily on the reader being familiar with CCNN

*4.2 Post-hoc Explanation*
-  It would be nice if the intuition behind Equation 6 was explained the same way it was for Equation 4

*L170: "We create under-represented regions by removing some species in the test split of [28] from each family in the train dataset."*

It should be made clear that the purpose of this is to do something like zero-shot learning.  Otherwise, this experiment looks like one that is clearly designed for this method to work.


*Table 5*

This caption should be explicit about the fact that these are classes where the training data was deliberately removed.

# Significance

If the concerns over hyper-parameter selection for TinyImageNet and why the Part/Meta-based augmentation strategies were excluded from Table 2 are addressed, this does seem like a significant contribution.

# Update based on discussion

The reviewer's main concerns have been addressed and the rating has been revised upwards by a point.

**Time Spent Reviewing:**

2.5

---

> ### Author Response · Authors · 2021-08-10
> **Response for Reviewer 4Jog**
>
> Thank you for your insightful comments.
>
> 1. **"We apply a softmax function to obtain the percentage contributions and select the set of concepts whose contribution is greater than some threshold." How is this threshold chosen?**
>
>    Given the number of concepts n in the concept layer of CCNN, the threshold is set to be 1/n.
>
>
>
> 2. **"We consider the image patches whose pixels have importance scores greater than some threshold as the concepts that have contributed to the model decision." How is this threshold chosen?**
>
>     The importance scores were normalized to have values between 0 and 1. The image patches whose pixels have scores greater than 0.5 were considered as the concepts that have contributed to the model decision.
>
>
>
> 3. **How were the hyper-parameters for TinyImagenet tuned?**
>
>     For TinyImageNet, we divide the training dataset into two sets. The first set has 400 images per class which we use for training our model. The second set has 100 images per class which we use for tuning the hyper-parameters.
>
>
>
> 4. **Report the results for WS-DAN and Metaset-based methods for the experiment using explanations from CCNN, the self-explainable model?**
>
>     We have carried out experiments to include comparison with methods (WS-DAN and Metaset-based) for the other external source data augmentation methods as shown the table below.
> | Method | CUB | CUB-Families  |
> | ------------- |:-------------:| ------------:|
> | Original dataset | 84.3 | 83.8 |
> | WS-DAN | 81.6 | 81.8 |
> | Metaset-based | 85.3 | 88.1 |
> | BRACE | 86.0 | 88.7 |
>
>
>
> 5. **Why are the results for selection methods based on ResNet101/DenseNet161 while the previous sections' results are based on ResNet34?**
>
>    “Meta-set based” method has been implemented on ResNet34. For a fair comparison, we also use
> ResNet34 with existing data augmentation methods.
>
>     We use different architectures ResNet101/DenseNet161 in the comparison with other selection methods to demonstrate that BRACE can also be applied to classification models with different architectures. Tables below show the results on ResNet34, the first table use post-hoc explanation and the second table use explanation from interpretable models. We see that BRACE remains the clear winner.
>
>    Comparison of classification accuracies using post-hoc explanation.
> | Method | CUB | CUB-Families  |
> | ------------- |:-------------:| ------------:|
> | Original dataset | 85.5 | 82.6 |
> | Random | 84.4 | 86.5 |
> | Confidence | 86.6 | 85.0 |
> | BRACE(utility) | 87.7 | 90.0 |
>
>    Comparison of classification accuracies using explanations from interpretable classifier CCNN.
> | Method | CUB | CUB-Families  |
> | ------------- |:-------------:| ------------:|
> | Original dataset | 84.3 | 83.8 |
> | Random | 84.5 | 86.5 |
> | Confidence | 84.7 | 84.8 |
> | BRACE(utility) | 86.0 | 88.7 |
>
>
>
> 6. **Why GradCAM is considered as concept-based explanation method?**
>
>     GradCAM generates explanations by calculating the importance of high-level features (feature maps from the last convolutional layer) towards the model decision. Existing work such as [1, 2] have demonstrated that these high-level features correspond to semantic concepts (e.g., tier of an automobile, ears of dogs). Hence, we consider that the image regions highlighted by GradCAM cover semantic concepts.
>
>     [1] “Visualizing and understanding convolutional neural networks”, ECCV 2014.
>     [2] “Network dissection: Quantifying interpretability of deep visual representations”, CVPR 2017.
>
>
>
> 7. **Why GradCAM is used for the experiments instead of TCAV and ACE?**
>
>     There are practical difficulties in using methods such as TCAV to derive the concepts that cause the misclassifications. In TCAV, the user has to manually define a set of concepts and provide labeled examples for each concept.  We cannot use ACE as it is dependent on the importance scores generated by TCAV and requires 10-20 examples for each concept to determine the importantance of a concept that may not be feasible in practice.
>
>     [3] A. Ghorbani, J.Wexler, J. Y. Zou, and B. Kim, “Towards automatic concept-based explanations,” in NeurIPS, 2019.
>     [4] B. Kim, M. Wattenberg, J. Gilmer, C. Cai, J. Wexler, F. Viegas, et al., “Interpretability beyond 328 feature attribution: Quantitative testing with concept activation vectors (tcav),” in ICML, 2018.

---

> > ### Comment · Reviewer_4Jog · 2021-08-10
> > **Response to the authors**
> >
> > 1-5, 7:  Thank you for the additional details and clarifications!
> >
> > 6.  Thank you for the references.  This still seems a little odd as GradCAM explanations are usually visualized on top of the images themselves (thus preventing them from directly conveying concepts) and it does seem a bit tricky to determine what "semantic concepts" are encoded in a neural network's representation.  But this does not seem to be a major concern.

---

### Official Review · Reviewer_zD5d · 2021-07-13

**Rating:** 6
**Confidence:** 4

**Summary:**

This paper proposed a utility function to rank new training samples with respect to their potential contributions of improving the model performance. The proposed utility contains two parts, one is for the under-represented region, another is based on explanation algorithms for finding the concepts that cause the misclassification. Experiments proved the effectiveness of the proposed method.

**Limitations And Societal Impact:**

.

**Main Review:**

Overall I find the idea of this paper interesting: using explanations that indicate the reasons of misclassification to improve the model performance is interesting and practically useful.

However, there are two main issues:

1. The sample selection from this paper is on web datasets, while authors only compared with two trivial methods, Random and Confidence. Though I am not familiar with the sample selection, I believe there exists some other more advanced baselines and I suggest authors compare with advanced baselines. Because, fundamentally, the proposed approach is for sample selection from an open dataset, instead of data augmentation, according to my understanding. Related key words may be learning from web images (found in the cited paper [29]), but others words may be also pertinent. I hope authors could address this issue with clear explanations and/or experiments.

2. The presentation is not good. (1) The samples in Figure 1 are mis-labeled by the dataset or mis-classified by the model? I didn’t understand the presentation of Figure 1. (2) Better highlight the definition of under-represented sample/region, or mention it at beginning (Line 89). (3) The designed scores should be better explained, with intuitions and remarks. For example, which samples will obtain high utility scores, and in which way these samples will help the model to generalize better. Without some intuitive introductions, the formulas are difficult to understand. (4) The designed formulas are strange and there are no explanations. For example, why exponential is needed in eq.(2)? Why log is needed in eq. (6)? Will some other alternatives be also useful? e.g., self-entropy as utility?

and several detailed questions:

1. How many samples have been selected for each dataset? I have seen the reported numbers for two methods. I believe the number of samples used by this paper is much smaller.
2. I briefly looked at the cited paper [29], which is a weakly supervised method. Authors used the collected samples from [29] for trainings on CUB. How the labels of these new samples are obtained? Or did I miss something?
3. Are the compared methods using samples from the same image repositories?



[29] C. Zhang, Y. Yao, H. Liu, G.-S. Xie, X. Shu, T. Zhou, Z. Zhang, F. Shen, and Z. Tang, “Web supervised network with softly update-drop training for fine-grained visual classification,” in AAAI, 2020.

**Time Spent Reviewing:**

3

---

> ### Author Response · Authors · 2021-08-10
> **Response to Reviewer zD5d**
>
> Thank you for your insightful comments.
>
> 1. **Compare with advanced sample selection baselines. Address this issue with clear explanations and/or experiments.**
>
>    We have carried out comparison with advanced sample selection methods. Specifically, we compare BRACE with two recent active learning methods, Coreset [1] and Learning Loss [2]. The following table shows that BRACE outperforms these methods.
>
> | Method | CUB | CUB-Families  |
> | ------------- |:-------------:| ------------:|
> | Original dataset | 87.4 | 85.4 |
> | Coreset | 85.9 | 88.7 |
> | Learning Loss | 84.8 | 88.6 |
> | BRACE (utility)| 89.2 | 91.2 |
>
>    [1] Ozan Sener and Silvio Savarese. Active learning for convolutional neural networks: A core-set approach, ICLR, 2018.
>    [2] Donggeun Yoo and In So Kweon. Learning loss for active learning, CVPR, 2019
>
>
>
> 2. **The samples in Figure 1 are mis-labelled by the dataset or mis-classified by the model?**
>
>    The samples in Figure 1 are mis-classified by the model. Figure 1 shows a case where the classifier has failed to learn the correct decision boundary probably because the training dataset contains a very few images of juvenile Black Tern. Since juvenile Black Tern and White Breasted Nuthatch look similar, the model might need more samples to learn the correct decision boundary.
>
>
>
> 3. **The designed scores should be better explained. Which samples will obtain high utility scores? Which way the samples selected by the designed utility score will help the model to generalize better?**
>
>    The samples representing missing variations of a class will obtain high utility scores, especially the samples containing concepts that caused the model to make misclassifications. Adding such samples increase the representativeness of the dataset. A model learned from a more representative dataset can generalize better. For instance, such a model can identify a wide variety of instances of a class as we have demonstrated in Section 5.4.
>
>
>
> 4. **Why exponential is needed in eq. (2)?**
>
>    Using exponential in Eqn (2) enables us to better differentiate samples that contain visual features similar to the existing samples from a class c, but the model misclassifies with higher confidence.
>
>
>
> 5. **Why log is needed in eq. (6)?**
>
>    Using negative log allows us to select an image that we are confident that it contains at least one concept that causes the misclassification instead of an image that contains multiple concepts, all with low confidence.
>
>
> 6. **How many samples have been selected for each dataset?**
>
>    The number of images added by each method for each dataset is shown below.
>
> | Method | CUB | CUB-Families  | TinyImagenet |
> | ------------ |:-----------:| ----------:| ----------:|
> | Metaset-based | 16000 | 16000 | 30000|
> | Random | 7000 | 2000 | 13000|
> | Confidence | 7000 | 2000 | 13000|
> | BRACE (utility)| 7000 | 2000 | 13000|
>
>
>
> 7. **How the labels of samples collected by [29] are obtained?**
>
>   Zhang et al. [29] use the name of a class as the query to collect the freely available images from a public website (e.g., Flickr) for that class. The retrieved images are considered to be from the class used in the query.
>
> [29] C. Zhang, Y. Yao, H. Liu, G.-S. Xie, X. Shu, T. Zhou, Z. Zhang, F. Shen, and Z. Tang, “Web supervised network with softly update-drop training for fine-grained visual classification,” in AAAI, 2020.
>
>
> 8. **Are the compared methods using samples from the same image repositories?**
>
>    Yes.

---

> > ### Comment · Reviewer_zD5d · 2021-08-18
> > **More concerns**
> >
> > Thanks authors' responses. I have several additional questions after reading the responses, and I hope authors could address my concerns below.
> >
> > > 1. Compare with advanced sample selection baselines. Address this issue with clear explanations and/or experiments.
> >
> > As I stated before, I am not familiar with the sample selection. But I found it strange that recent algorithms Coreset and Learning Loss (85.9, 84.8) did not provide better performance than Random (86.4) or Confidence (87.0) on CUB, and marginally better than Random (88.3) on CUB-families. Are their methods not suitable on these datasets? Or there are other reasons?
> >
> > > 2. Figure 1
> >
> > I got the point. However, I am not very comfortable to agree with this illustration. According to my comprehension, authors would like to use Figure 1 to say that juvenile and adult *Black Terns* are quite different, juvenile ones are more similar to *White Breasted Nuthatch* but there are only 3 out of 30 *Black Terns* in the dataset are juvenile ones. So the trained model on this dataset would easily misclassify the juvenile *Black Terns*. However, if authors augment the dataset with samples from juvenile *Black Terns*, then the decision boundary would move towards *White Breasted Nuthatch*. Will *White Breasted Nuthatch* be misclassified as *Black Terns* after the augmentation? Since they are very similar.
> >
> > I do agree with the motivation but that is my concern of using this illustration to present the motivation.
> >
> > > 4. eq(2) & 5. eq(6)
> >
> > I meant that the exponential (of a probability score) or the log (with *max* outside) operation does not change the final results. Since the samples are selected by their rankings, the scores will not affect the final rankings with or without the exponential or the log operations. What the specific reasons that authors use them?
> >
> > Others previous responses seem good to me. While I have one more question.
> >
> > 9.  The networks used in this paper are all based on CCNN? Except on Tiny ImageNet? If so, do BRACE benefit much more from attributions, than from image patches, according to the results on CUB, CUB-families and Tiny ImageNet? If not, do BRACE have some limitations on so-called well-represented datasets (showing only marginal improvements on Tiny ImageNet)?

---

> > > ### Author Response · Authors · 2021-08-20
> > > **Responses to new questions**
> > >
> > > Please find responses to your new questions bellow.
> > >
> > > 1. **I found it strange that recent algorithms Coreset and Learning Loss (85.9, 84.8) did not provide better performance than Random (86.4) or Confidence (87.0) on CUB, and marginally better than Random (88.3) on CUB-families. Are their methods not suitable on these datasets? Or there are other reasons?**
> > >
> > >    Learning Loss (LLoss) selects images that are predicted to have high losses and Coreset selects images such that the diversity of the dataset is increased. In our experiment, we use image sets obtained from online resources such as Flickr which may be noisy.
> > > We calculate the percentage of noisy images selected by each method in the top 1000 images. The results are shown in the table below.
> > >
> > >    | LLoss | Coreset | Random | Confidence |
> > > | -------- |:--------:| :----------:| :-----------:|
> > > |    88.5  |  21.0    |    17.6     |	       0        |
> > >
> > >    We see that both LLoss and Coreset tend to select more noisy images as compared to Random and Confidence leading to lower performance. Sample images selected by each method for two classes of CUB dataset can be found at: https://bit.ly/37TQp7o.
> > >
> > >
> > > 2. **Regarding Figure 1, I got the point. However, I am not very comfortable to agree with this illustration. According to my comprehension, authors would like to use Figure 1 to say that juvenile and adult Black Terns are quite different, juvenile ones are more similar to White Breasted Nuthatch but there are only 3 out of 30 Black Terns in the dataset are juvenile ones. So the trained model on this dataset would easily misclassify the juvenile Black Terns. However, if authors augment the dataset with samples from juvenile Black Terns, then the decision boundary would move towards White Breasted Nuthatch. Will White Breasted Nuthatch be misclassified as Black Terns after the augmentation? Since they are very similar.**
> > >
> > >    No, with the augmented dataset the model will learn a new set of discriminative features for Black Terns and White Breasted Nuthatch classes such that it can correctly differentiate images of both classes.  This is demonstrated by the improvement in classification accuracy of both classes after applying BRACE as shown in the table below.
> > >
> > >    | Model    | Black Tern | White Breasted Nuthatch |
> > > | -------- |:--------:| :----------:|
> > > |    Original dataset  |  67.7    |    96.7     |
> > > |    BRACE (utility) |   83.3          |       100.0         |
> > >
> > > 3. **Regarding eq(2) & eq(6), I meant that the exponential (of a probability score) or the log (with max outside) operation does not change the final results. Since the samples are selected by their rankings, the scores will not affect the final rankings with or without the exponential or the log operations. What the specific reasons that authors use them?**
> > >
> > >     We use the $\log$ function in Eqn (6) so that, given a set of concepts caused misclassification, we can rank an image higher if it has at least one concept with high confidence score over an image that contains multiple concepts, all with low confidence scores. Similarly, we use exponential function in Eqn (2) to select samples that contain visual features that are similar to the existing samples from a class c, but the model misclassifies with high confidence. We will illustrate this.
> > >
> > >     **Example 1.** Suppose there are 3 concepts that caused misclassifications of class $c$. Consider two images $x_1$ and $x_2$ which have same $\beta$ values, and each image has three image patches. Table below gives the degree of match between the image patches and the concepts that caused misclassification:
> > >
> > >    | Image | Concept 1 | Concept 2 | Concept 3|
> > > | -------- |:------------:| :-----------:| :-----------:|
> > > | $x_1$  |  (0.2, 0.1, 0.1)|    (0.05, 0.2, 0.1)     | (0.3, 0.4, 0.9) |
> > > | $x_2$  | (0.5, 0.3, 0.2) |     (0.1, 0.4, 0.5)      | (0.5, 0.5, 0.3) |
> > >
> > >      We would want to select $x_1$ over $x_2$ since $x_1$ has an image patch that matches Concept 3 with a high probability of 0.9.
> > > Without using the log function, we have $∆(S_{c→\bar{c}},x)= \sum_{u ∈U} \max_{w ∈W}⁡(\frac{u.w}{‖u‖‖w‖})$.
> > > Then the score for $x_1$ would be 1.3 (= 0.2 + 0.2 + 0.9) and the score for $x_2$ would be 1.5 (= 0.5 + 0.5 + 0.5). In this case, $x_2$ will be selected.
> > >
> > >      Instead, we use the log function, $∆(S_{c→\bar{c}},x)= \sum_{u ∈U} \max_{w ∈W}⁡(- \log \left[1-\frac{u.w}{‖u‖‖w‖} + \epsilon \right] )$ . Then the score for $x_1$ would be 2.74 (= 0.22 + 0.22 + 2.30) and the score for $x_2$ would be 2.07 (= 0.69 + 0.69 + 0.69), enabling $x_1$ to be selected.
> > >
> > >      **Example 2.** Suppose two images $x_1$ and $x_2$ have the same $\Delta$ values. Suppose the similarity between images of class $c$ and $x_1$ is 0.6 and the model’s confidence that $x_1$ is from $ \bar{c}$ is 0.6.  Also, suppose the similarity between images of class $c$ and $x_2$ is 0.4 and the model’s confidence that $x_2$ is from $ \bar{c}$ is 0.95.
> > >
> > >      In this situation, we want to select $x_1$ because $x_2$ is possibly a noisy image as it has a low similarity with the images of $c$.
> > >
> > >      Without using exponential function, we have $\beta(x, c, \bar{c}) =  \frac{f_x  .f_c}{‖f_x ‖  ‖f_c ‖}  × P(\bar{c}|x) $. Then the score for $x_1$ would be 0.36 (= 0.6 x 0.6) and the score for $x_2$ would be 0.38 (= 0.4 x 0.95). In this case, $x_2$ will be selected.
> > >
> > >      With exponential function, we have $\beta(x, c, \bar{c}) =  \frac{f_x  .f_c}{‖f_x ‖  ‖f_c ‖ }  × e^{P(\bar{c}|x)} $. Then the score for $x_1$ would be 1.09 (= 0.6 x 1.82) and the score for $x_2$ would be 1.03 (= 0.4 x 2.58). In this case, $x_1$ will be selected.
> > >
> > >
> > > 4. **The networks used in this paper are all based on CCNN? Except on Tiny ImageNet? If so, do BRACE benefit much more from attributions, than from image patches, according to the results on CUB, CUB-families and Tiny ImageNet? If not, do BRACE have some limitations on so-called well-represented datasets (showing only marginal improvements on Tiny ImageNet)?**
> > >
> > >    No, only the experiments in which BRACE uses explanations from fully interpretable model are based on CCNN. Experiments in which BRACE uses post-hoc explanations (GradCAM) are based on standard ResNet and DenseNet models trained with cross-entropy loss. Results of these experiments are given in Table 1, Table 3, and Table 7.
> > > A classifier trained using a well-represented dataset will have limited improvement using BRACE, which is the case for all data augmentation methods. However, building such well-represented datasets is expensive and rarely exist.

---

### Official Review · Reviewer_Wy9k · 2021-07-15

**Rating:** 6
**Confidence:** 4

**Summary:**

The paper proposes to use interpretation to select effective instances for augmenting training data. A utility function is developed, where images with high scores mean that: 1) the images are helpful for better learning class-c data, 2) the images could significantly correct the drawbacks of the model M. The paper discusses two interpretation scenarios, interpretable models (GAP+linear model) and post-hoc interpretation. The experiment includes a comparison to two types of baseline methods, data augmentation methods and data selection methods. Ablation studies are also conducted to show the effectiveness of the two components in the utility function. Different parts in the current version are somehow fragmented.

**Limitations And Societal Impact:**

Please refer to the main review.

**Main Review:**

Originality: The paper proposes an interesting application of interpretation to improve models. The current approach mainly combines existing interpretation methods, but the overall design is reasonable and well-motivated.

Quality: In general, the proposed approach is technically sound. The core contribution is an interpretation-driven utility function with two terms. The first term encourages selecting instances for better learning class c which was not well learned in the model. The second term encourages selecting instances containing crucial concepts. In this way, interpretation is fully utilized. The effectiveness of the two terms is also supported in the ablation study.
Meanwhile, there are several factors that could be further clarified or improved. First, why do we need NLP-based content for interpretation? Second, what is the running time of the approach? Third, the proposed method is more like an active learning method than data augmentation, but the baseline methods are not balanced to reflect this (i.e., typical methods for active learning or data selection are not included as baseline methods).

Clarity: The paper is, in general, well written and well organized. Figure 1 is also helpful for understanding the motivation. However, Section 4.2 could be extended with more details, especially for the last paragraph.

Significance: The proposed work should be interesting to XAI researchers. A potential limitation is that the interpretable model used in Section 4.1 is not a commonly used model in practice, which may limit the applicability of the proposed method. However, post-hoc methods, which are more flexible, are not fully discussed in this work.

**Time Spent Reviewing:**

2.5

---

> ### Author Response · Authors · 2021-08-10
> **Response for Reviewer Wy9k**
>
> Thank you for your insightful comments.
>
>
> 1. **Why do we need NLP-based content for interpretation?**
>
>    We chose CCNN and GradCAM to demonstrate that BRACE can be applied with different kinds of explanations, e.g., visual explanations as well as linguistic explanations.
>
>
>
> 2. **Comparison with typical methods for active learning or data selection.**
>
>    We have carried out an additional experiment to compare BRACE with two recent active learning methods, Coreset [1] and Learning
>  Loss [2]. The following table shows that BRACE outperforms these methods.
>
>   | Method | CUB | CUB-Families  |
>   | ------------- |:-------------:| ------------:|
>   | Original dataset | 87.4 | 85.4 |
>   | Coreset | 85.9 | 88.7 |
>   | Learning Loss | 84.8 | 88.6 |
>   | BRACE (utility)| 89.2 | 91.2 |
>
>    [1] Ozan Sener and Silvio Savarese. Active learning for convolutional neural networks: A core-set approach, ICLR, 2018.
>    [2] Donggeun Yoo and In So Kweon. Learning loss for active learning, CVPR, 2019
>
>
>
> 3. **A potential limitation is that the interpretable model used in Section 4.1 is not a commonly used model in practice, which may limit the applicability of the proposed method. However, post-hoc methods, which are more flexible, are not fully discussed in this work.**
>
>     We have designed the experiments to demonstrate the flexibility of BRACE. We show that BRACE is able to use explanations from interpretable classifiers (e.g., CCNN) as well as post-hoc explanations for the black-box models (e.g., GradCAM). Further, we also show that BRACE can be used with different types of explanations (visual and linguistic). Due to the page limit, we could only present experiments for CCNN and GradCAM but BRACE can be used with other inherently interpretable classifiers and post-hoc explanations methods that provide concept-based explanations.

---

### Official Review · Reviewer_NzM8 · 2021-07-16

**Rating:** 6
**Confidence:** 4

**Summary:**

This paper proposes an explanation-based data augmentation method for image datasets. Technically, this method defines a utility function to choose the data samples collected from the Internet. The utility function is defined to select the samples that could better reflect the under-represented region in the data distribution. The authors compare the proposed method with some existing data augmentation techniques on three datasets and showcase the effectiveness of the proposed method to helping achieve higher finetuning accuracies.

**Limitations And Societal Impact:**

This paper discusses one possible limitation in the conclusion section. I would appreciate it if the authors could include a more comprehensive discussion of the limitations and future works, for example, the scalability of the proposed method, the generalizability of the proposed method in other application domains, etc. Regarding the social impact, the authors could discuss the potential threats of the proposed method if attackers use it to generate poison data.

**Main Review:**

1. The paper mentioned two types of existing data augment methods: the first type of methods conduct certain transformations to input samples to explore the local regions around the given inputs. It is relatively straightforward to understand that these methods cannot explore the under-represented region because of the local exploration nature. The second type of methods collect new data from other sources and add those data into the training set. The paper first argues that these methods will introduce noise or OOD samples, which is again quite obvious. However, the authors then argue in the L59 these methods can neither explore the under-represented region. I am kind of doubting this argument. I could understand that the authors want to argue that these methods do not explicitly include a mechanism to explore the under-represented region. But this does not mean that they cannot achieve this goal. In fact, collecting diverse samples from other sources may also serve the purpose of exploring under-represented regions. I would suggest the authors provide some empirical supports on this argument other than the finetuning accuracy. For example, the authors could show the utility scores of samples selected by these methods to show whether they could uncover the under-represented region.

2. My second comment is that the first and second terms in the utility function may have a similar effort and select a similar set of samples. My hypothesis is that most misclassified samples are wrongly labeled because they contain the specific features that cause the model to make mistakes. To this end, It seems that the second term (explanation term) only helps to filter out the highly confusing samples that are caused by randomness. However, since the first terms select the samples that are most deviated from the class center of their true class. It is less likely that their misclassifications are caused by randomness. I appreciate that the authors provide an ablation study to showcase the effectiveness of each term. I would suggest the authors also report the difference in the sets of samples selected by each term and showcase some examples. IMHO, this will help better understand the different effects of the two terms and why the second term works better than the first one.

3. Regarding the proposed utility function, I found the following questions to be addressed to make it clearer.
(1) What is the GAP layer, and what is $o$_{i} in line 126? (2) For the post-hoc explanation models, why not directly using Grad-CAM to generate explanation? What is the insight of choosing R-CNN? Why not selecting other explanation methods? (3) In Eqn. (6), what is the insight of adding the first term (i.e., max_{w \in W}).

4. Regarding the evaluation, I have the following concerns: (1) Major concern: the performance improvement in most setups is marginal. I also checked the results with stds shown in the supplement and found out that in some cases, there are overlaps between the accuracies obtained by baselines and the proposed method. To better demonstrate the superiority of the proposed method, I would suggest the authors conduct some statistical testings (e.g., paired t-test) on the results; (2) I found some inconsistencies among the experiments. More specifically, I would suggest the authors comment what's the reason for only applying two baselines in the self-explainable model experiment. (3) As I mentioned earlier, besides the finetuning accuracy, I would also suggest reporting the results of other metrics to showcase the effectiveness of the proposed method (e.g., the utility scores of samples selected by each method and the sample overlapping rate in the ablation study, the error collection rate (the number of originally errors that are corrected after finetuning) and the number of newly introduced errors after finetuning). (4) I would suggest adding a hyper-parameter sensitivity test on \epsilon and the explanation method used in the black-box case.

5. I would suggest the authors run a grammar check on the current version. I found some typos and grammar errors (e.g., Line 141 convepts -> concepts).

==== After rebuttal. ====
The rebuttal responses addressed most of my comments (especially the potential overlap between the $\beta$ and $\Delta$ and the statistical tests). With that, I am happy to raise my score to above the acceptance threshold. I would further appreciate that if the authors could include the new experiments and other parts of the responses in the revised paper (Or the camera-ready version if get accepted).



**Time Spent Reviewing:**

about 7 hours

---

> ### Author Response · Authors · 2021-08-10
> **Response for Reviewer NzM8**
>
> Thank you for your insightful comments.
>
> 1. **Provide some empirical supports for the argument that existing augmentation methods that use images from external sources may not explore the under-represented regions. For example, show the utility scores of samples selected by these methods to show whether they could uncover the under-represented region.**
>
>    We have conducted an additional experiment to support the argument that existing augmentation methods that use images from external sources may not explore the under-represented regions. We calculated the utility score for the top-500 images selected by each method. For Meta-set-based method, we select a random subset of remaining images after removing out-of-distribution images. The following table
> shows the average utility score of the selected images. We observe that the average utility scores for baseline methods are significantly lower compared BRACE indicating that they have not been able to explore under-represented regions.
>
>    |  |Meta-set-based | Random | Confidence | BRACE|
>    | ------ |:--------:| ------:|-------:|-------:|
>    | CUB	| 0.43	| 0.24	|  0.69 | 5.07 |
>    | CUB-Families |	 3.25 | 0.97	| 0.42| 73.59 |
>
>
>
> 2. **What is the GAP layer, and what is o{i} in line 126?**
>
>     GAP layer refers to Global Average Pooling layer. o{i} is the output of {i}th node in the GAP layer of CCNN given an image x and it corresponds to the availability of the concept {i} in x.
>
>
>
> 3. **For the post-hoc explanation models, why not directly using Grad-CAM to generate explanation? What is the insight of choosing R-CNN? Why not selecting other explanation methods?**
>
>     We have used the explanations generated by Grad-CAM. However, we need to calculate the degree of match between the concepts extracted from Grad-CAM’s explanations with the images obtained from online repositories. As such, we use R-CNN to obtain the regions in these images that correspond to the semantic concepts in the GradCAM’s explanation.
>
>
> 4. **In Eqn. (6), what is the insight of adding the first term (i.e., max{w \in W})?**
>
>      \begin{equation}
>      z_u = \max_{w \in W} \left(-\log\left[1- \frac{u.w}{{\Vert u}\Vert \Vert w\Vert}  +\epsilon \right] \right)
>      \end{equation}
>
>     \begin{equation}
>     \Delta(\mathcal{S}_{c \rightarrow \bar{c}}, x) =  \sum_{u \in U}  z_u
>     \end{equation}
>
>
>    For each concept $u$ that causes the misclassification, $z_u$ indicates the degree of match between $u$ and some region $w$ in an
>    image obtained from the online repository.  We take the region with the maximum similarity as it indicates the highest probability of  $u$
>    being present in the image.
>
>    An image with higher $\Delta(\mathcal{S}_{c \rightarrow \bar{c}}, x)$ indicates that it contains many concepts that led the model to
>    misclassify. This image will help the model to learn correct decision boundaries and will be selected to augment the training dataset.
>
>
> 5. **Paired t-test on the results.**
>
>     We conducted a paired t-test on classification accuracy achieved by BRACE and each baseline method for different datasets. The null hypothesis is that BRACE and a given baseline method have the identical average classification accuracy for the given dataset.  Tables below show the p-values. We observe that p-values are consistently below 0.05 except for one case (comparison with WS-DAN on CUB) indicating that accuracy improvement achieved by BRACE is statistically significant.
>
>  P-values of the paired t-test on the results when post-hoc explanations are used.
>
> | Method | Original | Cut-mix | Snap-mix | WS-DAN | Metaset-based |
> | ------ |:--------:| ------:|---------:|--------:|------------:|
> | CUB	|0.008	|0.015	| 0.043 | 0.079	| 0.044|
> | CUB-Families |	0.002 |	0.008|	0.001 |	0.002 |	0.027|
> | TinyImageNet |	0.0004 |	0.0007|	0.041 |	0.039 |	0.0007 |
>
> P-values of the paired t-test on the results when explanations from interpretable model are used.
>
> | Method | Original | Cut-mix | Snap-mix |
> | ------ |:--------:| ------:|---------:|
> | CUB	|0.0007	|0.006	| 0.004 |
> | CUB-Families |	0.001 |	0.0005|	0.002 |
>
>
>
> 6. **Report the results for WS-DAN and Metaset-based methods for the experiment using explanations from CCNN, the self-explainable model?**
>
>     We have carried out experiments to include comparison with additional methods (WS-DAN and Metaset-based) for the self-explainable model as shown the table below.
>
> | Method | CUB | CUB-Families  |
> | ------------- |:-------------:| ------------:|
> | Original training dataset | 84.3 | 83.8 |
> | WS-DAN | 81.6 | 81.8 |
> | Metaset-based | 85.3 | 88.1 |
> | BRACE | 86.0 | 88.7 |
>
> 7. **Add a hyper-parameter sensitivity test on \epsilon.**
>
>       Epsilon is not a hyper-parameter. Instead, it is a small number added to eliminate the numerical instability due to the log function. We set epsilon to 1 x e -8.

---

> > ### Author Response · Authors · 2021-08-14
> > **Further analysis on ablation study**
> >
> > Here we answer the following question.
> >
> > **Report the difference in the sets of samples selected by each term and showcase some examples.**
> >
> > In selecting images to augment the train dataset, BRACE uses a utility function that evaluates whether a new image is in an under-represented region ($\beta$ in utility function) and whether it contains concepts that led to the misclassifications ($\Delta$ in utility function). We say that an image $x$ is an under-represented sample of class $c$ if it contains visual features of $c$ but the model's confidence that $x$ belongs to a different class $\bar{c} \neq c$  is high. $x$ can be an under-represented sample if $x$ is an image of a subcategory of $c$ that was absent in the dataset used to train the model, $x$ contains an unfamiliar background, $x$ does not show the discrminative features due to occlusions etc. When $\Delta$ is removed from the utility function (BRACE$^{-\Delta}$), images from the under-represented regions are selected but the selected images may not contain the concepts that caused misclassifications of $c$. On the other hand, when $\beta$ is removed from the utility function (BRACE$^{-\beta}$), the images containing concepts contributed for misclassifications are selected but those images may or may not fall in the under-represented regions.
> >
> > We demonstrate this behavior using the images selected by variants of BRACE for two classes of CUB-Families dataset, namely Alcedinidae and  Stercorariidae. Analyzing the explanations for misclassifications, we understand that the top-3 concepts that have caused most of the misclassifications in Alcedinidae class are *blue crown, white neck* and *brown crown*, while concepts *large wingspan, black crown* and *grey wing* have contributed for most of the misclassifications in Stercorariidae class. The images selected by each BRACE variant are shown in the file at - https://bit.ly/37I83Lb.
> >
> > Analyzing the selected images, we observe that BRACE$^{-\Delta}$ has selected images from under-represented regions. Either they belong to the subcategories that were excluded from the training dataset, have unfamiliar backgrounds or the discriminative features of the class are hidden due to poses or occlusions. However, the images selected by BRACE$^{-\Delta}$ do not include the concepts caused for misclassifications. On the other hand, BRACE$^{-\beta}$ selects images clearly showing one, or more concepts that have led to misclassifications, e.g., Alcedinidaes with *white neck* and Stercorariidae with *large wingspan*. However, some of the images selected by BRACE$^{-\beta}$ do not fall into under-represented regions. In contrast, BRACE has selected images from under-represented regions that contain concepts contributed for misclassifications. Such images are informative and help the model to learn correct boundaries, leading to higher accuracies.
> >
> >    Further, we calculate "sample overlapping rate" of images selected by different variants of BRACE. "Sample overlapping rate" refers to the intersection over union of the image sets selected by two variants. The sample overlapping rate between BRACE and BRACE$^{-\Delta}$ is 0.20 ,BRACE and BRACE$^{-\beta}$ is 0.37 and BRACE$^{-\Delta}$ and BRACE$^{-\beta}$ 0.09. These results indicate that when applied individually  $\beta$ and $\Delta$ select different image sets and $\Delta$ has a greater influence on the set of images selected by BRACE.

---

> > > ### Comment · Reviewer_NzM8 · 2021-08-16
> > > **Response to the authors**
> > >
> > > Thanks the authors for providing additional ablation studies. It makes sense to me now. Please correct me if my understanding is wrong. If we take DNN training as learning a branch of rules from the data. $\beta$ selects the samples with rules that DNN was not able to pick up during the training due to lack of samples. $\Delta$ represents the samples with wrong rules learned by the DNN. From this point of view, $\beta$ and $\Delta$ are different in that they select samples representing different sets of rules. Both types of samples could cause misclassifications but in different ways.

---

> > > > ### Author Response · Authors · 2021-08-17
> > > > **Response to new comments**
> > > >
> > > > Yes, this is correct.

---

> > ### Comment · Reviewer_NzM8 · 2021-08-16
> > **Response to the authors**
> >
> > Thanks the authors for responding to my comments and conducting the additional experiments. Most of the responses look good to me. Only two additional comments: (1) In the paired t-test, would the $H_{0}$ be "the proposed method has a lower average accuracy than the baselines" rather than "identical"? In this case, a low p-value means rejecting $H_{0}$, indicating the proposed method has a higher average accuracy than the baselines. (2) Maybe I missed it, but I didn't find a response to the point (3) in my comment 4 (evaluation comments). I would suggest the authors respond to it (even without additional experiments if time is limited).

---

> > > ### Author Response · Authors · 2021-08-17
> > > **Response to new comments**
> > >
> > > 1. **In the paired t-test, would Η_0  be "the proposed method has a lower average accuracy than the baselines" rather than "identical"? In this case, a low p-value means rejecting Η_0, indicating the proposed method has a higher average accuracy than the baselines.**
> > >
> > >    We have conducted a two-tailed t-test to test the null hypothesis that “BRACE has an average classification accuracy identical to that of a given baseline method for the given dataset.” In performing this test, we compute the t value where a positive value indicates that our proposed method has a higher average accuracy than baselines, while a negative value indicates otherwise. In our experiments, we have obtained positive t values that are greater than the threshold at 0.025 significance level, indicating that BRACE has achieved statistically significant higher average accuracy than baselines.
> > >
> > >
> > >
> > > 2. **Response to the point (3) in comment 4 (evaluation comments).**
> > >
> > >     The point (3) in comment 4 is
> > >
> > >     **Report the results of other metrics to showcase the effectiveness of the proposed method (e.g., the utility scores of samples selected by each method and the sample overlapping rate in the ablation study, the error collection rate (the number of originally errors that are corrected after finetuning) and the number of newly introduced errors after finetuning).**
> > >
> > >     We conducted additional experiments to compute the utility score for the top-500 images selected by each method. The following table shows the average utility score of the images. We observe that the average utility scores for baseline methods are significantly lower compared BRACE indicating that the images retrieved are not from the under-represented regions.
> > >
> > >     | | Meta-set-based | Random | Confidence | BRACE|
> > > | ------ |:--------:| ------:|-------:|-------:|
> > > | CUB	| 0.43	| 0.24	| 0.69 | 5.07 |
> > > | CUB-Families |	 3.25 | 0.97	| 0.42| 73.59 |
> > >
> > >     We have also calculated the "overlapping rate" of samples selected by different variants of BRACE in our ablation study. "Overlapping rate" refers to the intersection over union of the image sets selected by two variants. The overlapping rate between BRACE and BRACE$^{-\Delta} $ is 0.20, BRACE and BRACE$^{-\beta} $ is 0.37 and BRACE$^{-\Delta} $ and BRACE$^{-\beta} $ 0.09. These results indicate that when applied individually, $\beta$ and $\Delta$ will retrieve different image sets.
> > >
> > >     For other evaluation metrics, we are currently running experiments and will report the results when available.

---

> > > > ### Comment · Reviewer_NzM8 · 2021-08-17
> > > > **Raising the score**
> > > >
> > > > Thanks the authors for the further clarifications and additional experiments. Appreciated that. Overall, the rebuttal indeed addressed most of my comments. With that, I am happy to raise my score to above the acceptance threshold. I would further appreciate that if the authors could include the new experiments and other parts of the responses in the revised paper (Or the camera-ready version if get accepted).

---

> > > > > ### Author Response · Authors · 2021-08-18
> > > > > **Response to new comments**
> > > > >
> > > > > Thank you.
> > > > > We will certainly include the new experiments and other part of the responses in the revised version.

---

> > > > > > ### Comment · Reviewer_NzM8 · 2021-08-19
> > > > > > **Applications to other domains**
> > > > > >
> > > > > > Thanks for your commitment. Appreciate that. I actually really like the idea of using explanation to collect/generate more data for model patching/enhancement. An interesting future work (but definitely beyond the scope of this work) would be the extension of the proposed method to non-image domains, especially the domains where high-quality label data are not that easy to access (Medical, finance, etc). If we can somehow demonstrate that explanation could help reduce the labeling cost for those domains, it would also be a meaningful contribution. This is just my personal opinion. The authors are not required to respond to/address this comment. It won't influence my rating of this work. The current submission with the response is already in a good shape.

---

### Official Review · Reviewer_79xf · 2021-07-21

**Rating:** 7
**Confidence:** 4

**Summary:**

The paper proposes a data augmentation framework called BRACE that utilizes concept-based explanations from existing interpretability methods (such as Comprehensible Convolutional Neural Network (CCNN) and GradCAM) to identify informative examples to be added to the training set. Specifically, the proposed approach defines a utility function that focuses on finding images that are 1) under-represented in a class (i.e., images containing visual features of the class but are predicted as other classes with high confidence by the classifier), and 2) contain visual concepts that have led to the misclassification. Experiments on CUB, CUB-Families and Tiny Imagenet show better performance for the proposed approach as compared to previous works and baselines.

**Ethical Concerns:**

I do not think that there are any ethical concerns related to this work.

**Limitations And Societal Impact:**

The paper discusses their work’s societal impact satisfactorily. However, I think the paper can expand on the limitations a bit more, e.g., it might be difficult to derive the set of concepts that lead to misclassification for some types of interpretability methods, etc.

**Main Review:**

Strengths:
- The paper presents a very nice demonstration of how explainability can be used to improve image classification systems.
- The paper is very well written and easy to follow.
- The ablation studies clearly highlight the importance of both of the main components of the proposed framework.


Weaknesses:
- There seems to be some mismatch between the motivation and the proposed approach. In Line 87, the authors claim that the queried set of images is typically noisy but then their proposed solution focuses on under-representation and visual features that lead to misclassification. Shouldn’t the solution just be to throw out the noisy data? They seem to be orthogonal to each other. Please clarify.
- If there are two highly similar images in the queried set of images such that they both result in high utility scores, the proposed method would choose to add both to the training set, correct? This doesn’t seem ideal. Is there a way to enforce some diversity in the selected set of images to maximize the information gain?
- I appreciate the experiment showing better generalization with BRACE as compared to the baselines. But, how does BRACE compare against previous works in this experiment?
- The derivation of the set of concepts that caused the classifier to misclassify images of class c to c_bar as well as their overlap with the visual features in an image is specific to an interpretability method. I think it is okay to consider this issue as out of scope for this work but it would be good for the paper to discuss some initial ideas about how they can be generalized to a family of interpretability methods.
- Where are the feature vectors extracted from that are used in equation 2?
- Lines 64-66: How does GradCAM explain model decisions in terms of human-friendly concepts?
- The citation for the RCNN model in line 155 is incorrect.


-----------------------------------------------------------------------------------------------------------------------------------------------------------------------------------------
-----------------------------------------------------------------------------------------------------------------------------------------------------------------------------------------
I thank the authors for their response. It answers all my concerns.

After reading other reviews and the authors’ responses to them, I think all the major concerns from the other reviewers have also been answered.

In my opinion, the paper proposes a significant contribution that would be useful to the Explainable AI (XAI) community and, therefore, I would like to see the paper getting accepted (hence, keeping my acceptance rating).

I would encourage the authors to incorporate the additional experiments and the qualitative results they provided in the rebuttal period into the revised version (camera-ready, if accepted) of their paper.

**Time Spent Reviewing:**

4

---

> ### Author Response · Authors · 2021-08-10
> **Response for Reviewer 79xf**
>
> Thank you for your insightful comments.
>
> 1. **There seems to be some mismatch between the motivation and the proposed approach. In Line 87, the authors claim that the queried set of images is typically noisy but then their proposed solution focuses on under-representation and visual features that lead to misclassification. Shouldn’t the solution just be to throw out the noisy data? Please clarify.**
>
>    Our motivation is to learn the correct decision boundaries for the under-represented regions to improve classification accuracy. Our approach is to augment the training dataset with a set of samples queried from external online repositories, which can be noisy. Simply throwing out the noisy data does not help as evident in the superior performance of BRACE over Metaset-based method. Instead, we design a new scoring function that differentiates samples in the under-represented with visual features that lead to the misclassification to enable the model to learn better decision boundaries.
>
>
> 2. **If there are two highly similar images in the queried set of images such that they both result in high utility scores, the proposed method would choose to add both to the training set? Is there a way to enforce some diversity in the selected set of images to maximize the information gain?**
>
>     We have included a pre-processing step to remove images that are similar to the ones in the training image set as well as duplicates obtained from the online repositories. This would ensure no two images are highly similar in the queried set of images, thus enforcing some diversity in the selected set of images.
>
>
> 3. **How does BRACE compare against previous works in the experiment demonstrating better generalizations of BRACE?**
>       We have conducted additional experiments to show the performances of previous works on NAbirds-Sub and ImageNet-V2-Sub. Both datasets have not been used in the training of the models. The results in the table below show that BRACE is able to achieve higher classification accuracy compared to existing data augmentation methods on these two datasets, demonstrating that BRACE can generalize better.
> | Method | NAbirds-Sub | ImageNet-V2-Sub  |
> | ------------- |:-------------:| ------------:|
> | Original training dataset      | 81.5 | 54.35 |
> | Cut-mix      |   72.1   |  37.2   |
> | Snap-mix |  76.0    |  42.6   |
> | WS-DAN | 67.0 | 56.1|
> | Metaset-based | 83.5 | 42.4 |
> | BRACE | 84.9 | 70.0 |
>
>
> 4. **How BRACE can be generalized to a family of interpretability methods?**
>     The proposed approach to use GradCAM with BRACE can be applied to any explanation method that provides super-pixel-based explanations, e.g., ACE [1] and Interpretable Basis Decomposition [2].
>
>     [1] A. Ghorbani, J.Wexler, J. Y. Zou, and B. Kim, “Towards automatic concept-based explanations,” in NeurIPS, 2019.
>     [2] B. Zhou, Y. Sun, D. Bau, and A. Torralba, “Interpretable basis decomposition for visual explanation,” in ECCV, 2018.
>
>
> 5. **Where are the feature vectors extracted from that are used in equation 2?**
>
>     Feature vectors are extracted from the classification model being explained (M).
>
>
> 6. **How does GradCAM explain model decisions in terms of concepts?**
>
>     GradCAM generates explanations by calculating the importance of high-level features (feature maps from the last convolutional layer) towards the model decision. Existing work such as [3, 4] have demonstrated that these high-level features correspond to semantic concepts (e.g., tier of an automobile, ears of dogs). Hence, we consider that the image regions highlighted by GradCAM cover semantic concepts.
>
>     [3] “Visualizing and understanding convolutional neural networks”, ECCV 2014.
>     [4] “Network dissection: Quantifying interpretability of deep visual representations”, CVPR 2017.

---

### Decision · Program_Chairs · 2021-09-27

**Decision:**

Accept (Poster)

**Comment:**

This paper has novelty as it proposes a form of data augmentation that uses explanations to select external images from poorly performing classes. All five reviewers gave a 6 or 7 rating. The authors provided detail responses to reviewers' comments. The reviewers and AC reached the agreement to accept the paper.